# QTALE: Quantization-Robust Token-Adaptive Layer Execution for LLMs

Kanghyun Noh [1]   Jinheon Choi [2]   Yulhwa Kim [3]

## Abstract

Large language models (LLMs) demand substantial computational and memory resources, posing challenges for efficient deployment. Two complementary approaches have emerged to address these issues: token-adaptive layer execution, which reduces floating-point operations (FLOPs) by selectively bypassing layers, and quantization, which lowers memory footprint by reducing weight precision. However, naively integrating these techniques leads to additional accuracy degradation due to reduced redundancy in token-adaptive models. We propose QTALE (Quantization-Robust Token-Adaptive Layer Execution for LLMs), a novel framework that enables seamless integration of token-adaptive execution with quantization while preserving accuracy. Conventional token-adaptive methods reduce redundancy in two ways: (1) by limiting the diversity of training paths explored during fine-tuning, and (2) by lowering the number of parameters actively involved in inference. To overcome these limitations, QTALE introduces two key components: (1) a training strategy that ensures diverse execution paths are actively explored during fine-tuning, and (2) a post-training mechanism that allows flexible adjustment of the execution ratio at inference to reintroduce redundancy when needed. Experimental results show that QTALE enables seamless integration of token-adaptive layer execution with quantization, while keeping the accuracy gap to quantization-only models below 0.5% on CommonsenseQA benchmarks. By combining token-adaptive execution for FLOPs reduction and quantization for memory savings, QTALE provides an effective solution for efficient LLM deployment.

## 1. Introduction

LLMs have demonstrated remarkable proficiency in a wide range of natural language processing tasks (Zhang et al., 2022; Touvron et al., 2023; Grattafiori et al., 2024; Yang et al., 2025). Consequently, they have become the core components of modern AI applications. However, the substantial size of these models poses significant challenges for real-world deployment. In particular, their high memory consumption and computational demands substantially increase inference cost and latency, limiting accessibility and scalability in resource-constrained environments. These constraints hinder the widespread adoption of LLMs. As a result, improving the efficiency of LLM inference has become a central research direction, with efforts focused on reducing computational cost and memory footprint while maintaining task accuracy.

Recent research has introduced several techniques for efficient LLM inference, such as pruning (Frantar & Alistarh, 2023; Sun et al., 2024; Song et al., 2024), quantization (Frantar et al., 2023; Dettmers et al., 2022; Zhang et al., 2024), and token-adaptive execution (Jiang et al., 2024; Liu et al., 2023). Each of these methods exploits redundancy in large models but targets different efficiency dimensions: quantization reduces memory footprint by lowering weight precision, while token-adaptive layer execution reduces FLOPs by bypassing unimportant layers. Despite their complementary benefits, these techniques are typically studied in isolation. When applied together, their naive integration often leads to additional accuracy degradation due to compounded redundancy reduction. This creates a critical need for a unified approach that combines the strengths of both techniques while mitigating their drawbacks.

In this paper, we propose QTALE, a novel framework that seamlessly integrates token-adaptive layer execution with quantization while preserving accuracy. QTALE addresses the key limitations of conventional token-adaptive methods, namely reduced training-path redundancy and reduced parameter redundancy, through two innovations:

- A quantization-robust training strategy that ensures diverse execution paths are explored during fine-tuning

- A post-training execution ratio adjustment mechanism that reintroduces redundancy at inference time to improve robustness against quantization errors.

---

[1]Department of Semiconductor Convergence Engineering, Sungkyunkwan University [2]Department of Electrical and Computer Engineering, Sungkyunkwan University [3]Department of Semiconductor Systems Engineering, Sungkyunkwan University. Correspondence to: Yulhwa Kim <yulhwa.kim@skku.edu>.

*Proceedings of the 43rd International Conference on Machine Learning*, Seoul, South Korea. PMLR 306, 2026. Copyright 2026 by the author(s).

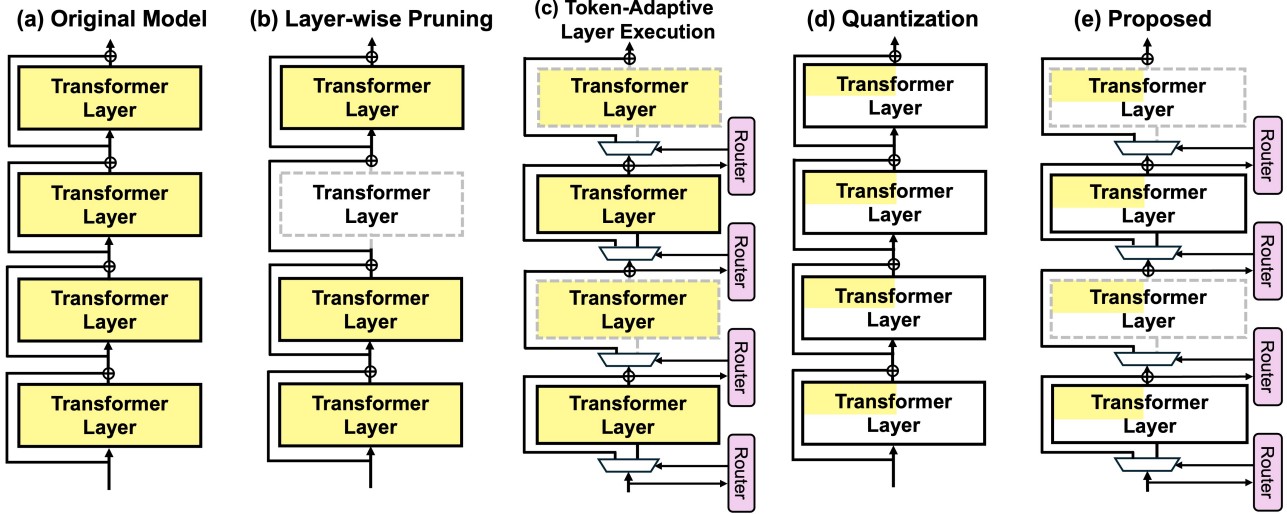

*Figure 1.* Overview of a standard LLM architecture and representative techniques for efficient inference. The fraction of color fill in each transformer layer denotes memory cost per layer, while dashed gray outlines indicate skipped execution.

Through these contributions, QTALE enables the effective integration of token-adaptive layer execution with quantization, thereby reducing both FLOPs and memory usage.

## 2. Background

### 2.1. Transformer Layer-wise Pruning

Recently, many studies have demonstrated that LLMs exhibit redundancy at the transformer layer level (Song et al., 2024; Men et al., 2025; Kim et al., 2024). During inference, consecutive transformer layers often produce highly similar outputs, since each block incrementally contributes to the residual stream that spans the entire network. As shown in Figure 1(a), Modern LLM architectures are typically built on residual connections, where the output of each transformer layer is the sum of the previous layer output and the current layer computation:

$$x_{l+1} = x_l + f_l(x_l) \tag{1}$$

where $x_l$ is the input to the $l$-th layer and $f_l(\cdot)$ is the transformer layer function. If $x_{l+1}$ is sufficiently similar to $x_l$, the removal of the $l$-th layer has little effect on the final prediction. As layer-wise pruning (Figure 1(b)) removes both the parameters of a layer and its associated computations, it reduces FLOPs and memory overhead proportionally to the number of pruned layers. However, because it eliminates entire transformer blocks, achieving high pruning ratios (e.g., beyond 20%) typically leads to significant accuracy degradation.

### 2.2. Token-Adaptive Layer Execution

LLMs exhibit contextual sparsity, where only a subset of computations is required to generate each token. Previous works on token-adaptive execution have leveraged this sparsity to improve inference efficiency (Hoefler et al., 2021; Schuster et al., 2022; Del Corro et al., 2023; Luo et al., 2025; He et al., 2025; Jaiswal et al., 2024; Jiang et al., 2024; Liu et al., 2023). Building on this idea, D-LLM (Jiang et al., 2024) integrates both layer-wise redundancy and contextual sparsity by applying token-adaptive execution at the transformer layer level, achieving significant FLOPs reduction while maintaining accuracy.

As shown in Figure 1(c), D-LLM introduces a router module $g_l$ for each transformer layer to decide whether to execute or bypass that layer. Each router is a lightweight Multi-Layer Perceptron (MLP) performing binary classification (execute or bypass). During inference, the router selects the class with the higher score:

$$b_l = \mathbb{1}(\arg\max(g_l(x_l))) \tag{2}$$

where $\mathbb{1}(\cdot)$ denotes the one-hot operation, and $b_l$ is a two-dimensional decision vector resulting in either $[1, 0]$ (execute layer) or $[0, 1]$ (bypass layer). The output of the $l$-th layer is then computed as:

$$x_{l+1} = \begin{cases} x_l + f_l(x_l), & \text{if } b_l = [1, 0] \\ x_l, & \text{if } b_l = [0, 1] \end{cases} \tag{3}$$

D-LLM trains both the router parameters and task-specific adapters (Hu et al., 2022) during fine-tuning to adapt pre-trained LLMs to downstream tasks under token-adaptive execution. Here, as the $\arg\max$ operation is non-differentiable and deterministic, D-LLM uses the Gumbel-Softmax with reparameterization trick and straight-through estimator for the training.

To achieve the target execution ratio, D-LLM introduces a ratio regularization loss $\mathcal{L}_{rate}$ and the overall training

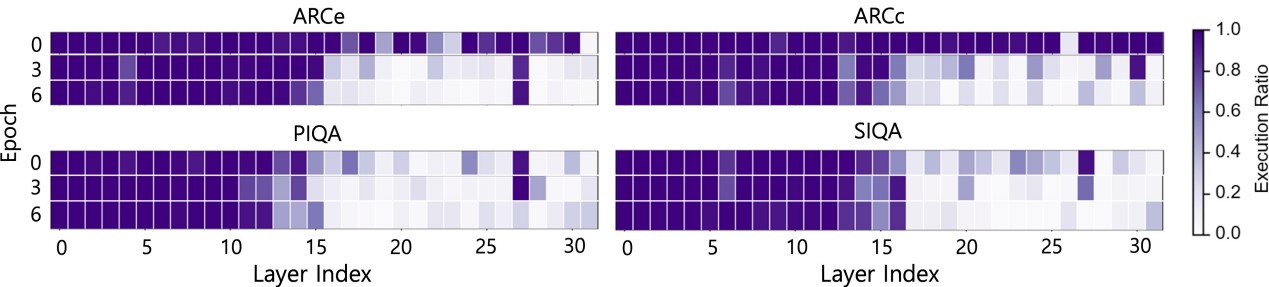

*Figure 2.* Heatmap of the average execution ratio for each layer of LLaMA3.1-8B with D-LLM. The ratios are measured on the first 200 training samples after fine-tuning epochs 0, 3, and 6, across four CommonsenseQA datasets: ARCe, ARCc, SIQA, and PIQA.

objective of D-LLM combines the cross-entropy loss $\mathcal{L}_{CE}$ with this regularization:

$$\mathcal{L}_{\text{D-LLM}} = \mathcal{L}_{CE} + \lambda_1 \cdot \mathcal{L}_{rate}$$
$$\text{s.t.} \quad \mathcal{L}_{rate} = |R_{avg} - R_{target}| \tag{4}$$

where $R_{avg}$ denotes the average execution ratio across all layers during inference, and $R_{target}$ is the desired target ratio. $\lambda_1$ is a hyperparameter that controls the strength of $\mathcal{L}_{rate}$. In D-LLM, $R_{target}$ is set to 0.5. After fine-tuning, D-LLM achieves the target execution ratio and reduces the FLOPs required for LLM inference to about 50% of those of the original model. Although token-adaptive execution can deliver substantially higher FLOPs reduction compared to layer-wise pruning, it leaves memory overhead unaddressed since the full set of model parameters remains stored. Hence, a complementary strategy is necessary to reduce both computational cost and memory footprint simultaneously.

### 2.3. Quantization

Quantization is a widely adopted compression technique that reduces model size by lowering the precision of weight parameters from high to low precision (Dettmers et al., 2022; Xiao et al., 2023; Frantar et al., 2023; Lin et al., 2024). Recent studies show that weights can be quantized to 4-bit integers without significant accuracy loss when combined with careful calibration, even under post-training quantization (PTQ) (Lin et al., 2024; Zhang et al., 2024). Since conventional LLMs store weights in 16-bit floating-point (FP) format, 4-bit quantization achieves up to a $4\times$ reduction in model size and effectively alleviates memory overhead (Figure 1(d)). Therefore, modern PTQ algorithms such as AWQ (Lin et al., 2024) are integrated into widely used LLM serving frameworks (e.g., vLLM), further enhancing deployment practicality. However, quantization does not reduce FLOPs, as the total number of operations remains unchanged. Thus, integrating the two techniques (Figure 1(e)) offers the potential to build a more efficient LLM execution model that simultaneously addresses computational cost and memory footprint.

## 3. Proposed QTALE

### 3.1. Challenges of Integrating Token-Adaptive Execution with Quantization

While token-adaptive layer execution reduces FLOPs and quantization reduces memory overhead, directly applying PTQ to the token-adaptive execution model D-LLM introduces additional accuracy degradation (details are provided in the Experimental Section and Appendix A.3). This degradation arises from reduced redundancy in D-LLM models, which can be examined from two perspectives.

**First, reduced training-path redundancy.** Although D-LLM is designed for token-adaptive execution, its training objective focuses only on meeting the average target execution ratio. This allows solutions where half of the layers are permanently executed while the others are permanently bypassed. Consequently, as shown in Figure 2, instead of evenly distributing execution across layers, D-LLM often converges to highly uneven execution patterns, closely resembling layer-wise pruning. At the start of fine-tuning, router modules are biased toward execution, so most layers are active. As training progresses, certain layers gradually stop receiving execution signals and thus rarely participate in training. For example, in LLaMA3.1-8B, the 20th, 23rd, and 26th layers receive less than 5% execution ratio after fine-tuning, with their ratios dropping sharply within the first three epochs of a 10-epoch training process (Figure 3a). As a result, these layers have little opportunity to participate in fine-tuning. This leads to sparsely explored paths through the model, ultimately limiting robustness.

**Second, reduced parameter redundancy.** Deep learning models are generally overparameterized to enhance training capacity, making them inherently tolerant to moderate errors during inference (Allen-Zhu et al., 2019). For example, when a large pre-trained model is quantized, the network can rely on redundant parameters to absorb quantization errors and preserve accuracy. In contrast, D-LLM achieves efficiency by processing only about half of the transformer layers. As a result, each parameter becomes more critical to

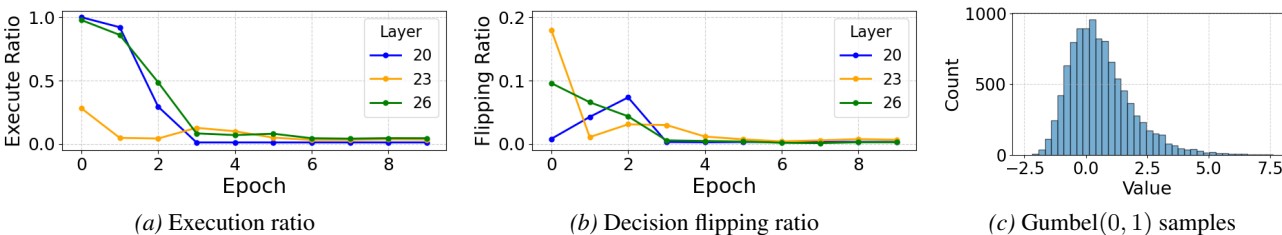

*(a)* Execution ratio    *(b)* Decision flipping ratio    *(c)* Gumbel$(0, 1)$ samples

*Figure 3.* Execution behavior of D-LLM. (a) Execution ratio, (b) execution decision flipping induced by Gumbel noise across fine-tuning epochs, and (c) histogram of samples from $\pi \sim \text{Gumbel}(0, 1)$. Results are shown for layers 20, 23, and 26 of LLaMA3.1-8B on ARCe.

inference, and quantization errors have a disproportionately large impact on accuracy.

In summary, D-LLM reduces redundancy by both limiting the diversity of training paths and lowering the number of active parameters during inference. This reduction in redundancy makes the model less robust to quantization. Therefore, integrating token-adaptive execution with quantization requires careful management of redundancy to preserve overall model robustness.

### 3.2. Overview of QTALE

We propose QTALE, a token-adaptive execution method designed to be resilient against quantization errors, thereby enabling seamless integration with quantization without sacrificing accuracy. To address the two key limitations of conventional token-adaptive methods, namely reduced training path redundancy and reduced parameter redundancy, QTALE introduces two components: (1) a novel training strategy that involves diverse execution paths during fine-tuning and (2) a post-training mechanism for adjusting the execution ratio at inference, providing flexible control over redundancy.

### 3.3. Quantization-Robust Training for Token-Adaptive Execution

According to transformer layer-wise pruning studies (Song et al., 2024; Men et al., 2025; Kim et al., 2024), LLMs contain layers with relatively low contributions to residual path propagation. As a result, uneven execution ratios that deactivate certain layers are consistent with the inherent characteristics of LLMs, since not every layer contributes equally to final model performance. However, if execution decisions consistently favor a fixed subset of layers, large portions of the model remain under-trained, reducing redundancy and limiting robustness. To address this, we introduce randomness in path generation to enhance training-path redundancy. This idea is inspired by stochastic regularization techniques such as dropout (Srivastava et al., 2014) and stochastic depth (Huang et al., 2016), which improve generalization by randomly dropping neurons or entire layers during training. In a similar vein, introducing controlled

randomness into execution decisions forces different subsets of layers to participate in training, ensuring that more paths are explored.

As discussed in Section 2.2, D-LLM uses Gumbel-Softmax instead of $\arg\max$ in Eq. 2 during training. In this approach, the forward pass uses a hard mode of Gumbel-Softmax:

$$\hat{b}_l = \mathbb{1}(\arg\max(\log(\hat{g}_l(x_l)) + \pi)), \quad \pi \sim \text{Gumbel}(0, 1) \tag{5}$$

where $\pi$ is noise sampled from a Gumbel distribution. Please note that while the logarithm notation $\log(\hat{g}_l(x_l))$ is often used in the Gumbel-Softmax equation, in practice the operation directly accepts logits, which is the router output $g_l(x_l)$ in this case. During backpropagation, a soft mode is applied:

$$\tilde{b}_{l,i} = \frac{\exp\left((\log(\hat{g}_l(x_l))_i + \pi_i)/\tau\right)}{\sum_i \exp\left((\log(\hat{g}_l(x_l))_i + \pi_i)/\tau\right)}, \quad i \in \{0, 1\} \tag{6}$$

where $\tau$ is a temperature parameter that controls the sharpness of the softmax. Since this approach introduces Gumbel noise $\pi$, it initially injects stochasticity into routing decisions. However, during D-LLM training, this stochastic effect gradually diminishes. Because the distribution of $\pi \sim \text{Gumbel}(0, 1)$ is centered near 0 (Figure 3c), the router logits must remain within a moderate range (e.g., approximately $[-1, 1]$) for the noise to effectively flip decisions and introduce stochasticity. Yet, as the training objective of D-LLM focuses solely on maximizing accuracy and meeting the target execution ratio, the gaps between bypass and execute logits grow progressively larger as training advances (Figure 4a). In this regime, the influence of Gumbel noise becomes negligible, and decision flipping due to noise injection rarely occurs. For example, Figure 3b shows that the ratio of decision flipping caused by Gumbel noise drops to zero after Epoch 4, indicating that stochasticity is essentially lost.

If the gap between bypass logits and execute logits can be properly regulated, the Gumbel noise can effectively induce stochastic decisions, allowing diverse training paths and preventing the model from collapsing into a fixed execution pattern. To achieve this, we introduce an entropy

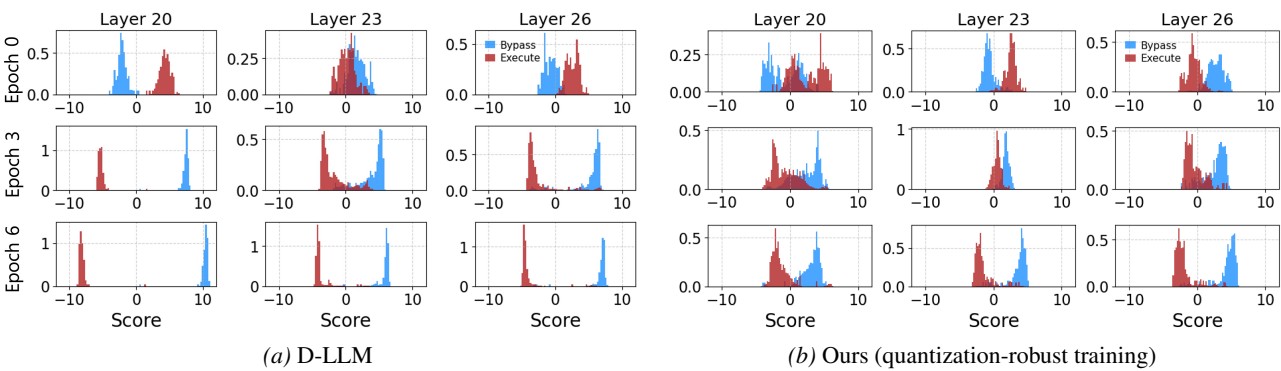

*(a)* D-LLM                                  *(b)* Ours (quantization-robust training)

*Figure 4.* Histograms of router output logits for three low-execution layers (20, 23, and 26) of LLaMA3.1-8B on ARCe. Logits are computed from the first 200 training samples after fine-tuning epochs 0, 3, and 6.

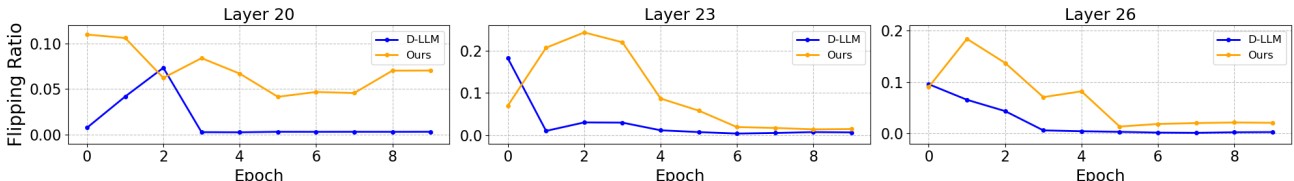

*Figure 5.* Comparison of Gumbel-noise–induced execution decision flipping across fine-tuning epochs between D-LLM and the proposed quantization-robust training. Results are shown for three low-execution layers (20, 23, and 26) of LLaMA3.1-8B on the ARCe dataset.

regularization loss on the router outputs:

$$\mathcal{L}_{entropy} = -\frac{1}{N_{layer}} \sum_{l=0}^{N_{layer}} \sum_{i=0}^{1} \tilde{b}_{l,i} \log(\tilde{b}_{l,i}) \quad (7)$$

where $N_{\text{layer}}$ is the total number of transformer layers, and $\tilde{b}_{l,i}$ denotes the soft probability of the $i$-th decision for layer $l$ (Eq. 6). A higher entropy corresponds to a smaller logit gap between bypass and execute classes, thereby increasing the likelihood that Gumbel noise can flip decisions and introduce stochasticity. By encouraging higher entropy during training, more diverse execution paths are explored, ensuring that additional layers remain actively involved in fine-tuning. The final fine-tuning objective is defined as:

$$\mathcal{L}_{\text{total}} = \mathcal{L}_{CE} + \lambda_1 \cdot \mathcal{L}_{rate} - \lambda_2 \cdot \mathcal{L}_{entropy} \quad (8)$$

The hyperparameter $\lambda_2$ balances the contribution of entropy maximization. By subtracting $\mathcal{L}_{entropy}$, the training process explicitly encourages higher entropy.

Figure 4b shows the histogram of router logits after training with the proposed quantization-robust method, whose training objective is defined in Eq. 8. Compared to the original D-LLM results in Figure 4a, the gap between bypass and execute logits is substantially narrower. As a result, the router outputs remain within a range where Gumbel noise can meaningfully influence routing decisions. This increases the likelihood of stochastic flipping in execution outcomes, as illustrated in Figure 5. Such stochastic path exploration prevents the model from over-relying on a small

subset of layers, ensures more balanced participation of layers during training, and ultimately enhances robustness to quantization.

### 3.4. Execution Ratio Adjustment Mechanism

Since token-adaptive layer execution inherently reduces parameter redundancy by activating only a subset of layers, slightly increasing the execution ratio can reintroduce sufficient redundancy to absorb quantization errors and better preserve accuracy. Although this adjustment introduces a modest increase in FLOPs, the resulting improvement in robustness to quantization enables seamless integration with quantization techniques. This integration reduces memory overhead and improves the overall efficiency of LLMs.

However, conventional D-LLM provides no mechanism for tuning the execution ratio at inference time. During inference, the execution decision for each layer is determined by an $\arg\max$ operation on the router output: a layer is executed if the score for execution exceeds the score for bypassing (Eq. 2). This rule locks the model to the execution ratio established during training, where the ratio is enforced through the regularization loss $\mathcal{L}_{rate}$ (Eq. 4). As a result, any adjustment to the target ratio requires retraining.

Retraining to achieve the redundancy needed for each deployment setting is impractical. Therefore, to design an execution mechanism with inference-stage adjustability, the router must include a tunable component. Moreover, to ensure predictable effects of such adjustments, this tunable

*Table 1.* Accuracy and PPL comparison on LLaMA2-7B and LLaMA3.1-8B. Accuracy is reported on CSQA and MMLU, while PPL is reported on Alpaca. (Avg.: average)

| Bits | Layer Execution | CSQA | | | | | | | | MMLU ($\uparrow$) | Alpaca ($\downarrow$) |
| | | PIQA | BoolQ | SIQA | ARCe | ARCc | Winogr. | OBQA | Avg. ($\uparrow$) | | |
|---|---|---|---|---|---|---|---|---|---|---|---|
| | | | | | **LLaMA2-7B** | | | | | | |
| | Full | 83.73 | 87.98 | 79.58 | 82.53 | 65.27 | 81.61 | 83.00 | 80.53 | 54.26 | 3.23 |
| 16 | D-LLM | 83.51 | 88.17 | 79.02 | 81.06 | 66.04 | 81.22 | 81.40 | 80.06 | 52.83 | 4.33 |
| | QTALE | 84.06 | 88.22 | 78.97 | 81.65 | 65.70 | 81.45 | 82.40 | **80.35** | **53.00** | **4.09** |
| | Full | 81.34 | 87.67 | 79.53 | 79.97 | 62.37 | 81.14 | 79.40 | 78.77 | 51.74 | 3.22 |
| 4 | D-LLM | 81.23 | 86.20 | 77.18 | 79.08 | 62.03 | 78.14 | 77.40 | 77.32 | 50.47 | 4.43 |
| | QTALE | 83.30 | 87.67 | 78.56 | 79.59 | 66.21 | 79.95 | 80.20 | **79.18** | **51.24** | **3.74** |
| | Full | 78.02 | 83.76 | 74.36 | 71.38 | 55.29 | 74.51 | 68.20 | 72.22 | 46.12 | 3.30 |
| 3 | D-LLM | 74.65 | 80.02 | 73.34 | 71.55 | 55.03 | 74.43 | 65.00 | 70.57 | 42.84 | 5.35 |
| | QTALE | 77.58 | 83.82 | 73.34 | 73.44 | 57.17 | 74.98 | 69.20 | **72.79** | **44.78** | **4.38** |
| | | | | | **LLaMA3.1-8B** | | | | | | |
| | Full | 80.04 | 88.19 | 88.90 | 87.58 | 77.03 | 84.21 | 85.20 | 81.28 | 59.12 | 3.57 |
| 16 | D-LLM | 79.84 | 86.02 | 89.35 | 86.24 | 75.68 | 83.43 | 84.20 | 80.45 | **58.85** | 5.06 |
| | QTALE | 78.81 | 86.18 | 87.37 | 87.16 | 78.16 | 83.43 | 84.80 | **80.54** | 58.40 | **4.90** |
| | Full | 79.53 | 86.29 | 87.52 | 86.07 | 75.17 | 83.58 | 83.00 | 79.67 | 56.16 | 3.65 |
| 4 | D-LLM | 79.27 | 83.57 | 86.88 | 84.93 | 69.88 | 80.51 | 81.00 | 77.68 | 55.36 | 5.47 |
| | QTALE | 79.27 | 85.26 | 88.13 | 85.56 | 74.83 | 82.08 | 82.40 | **79.17** | **55.86** | **4.11** |
| | Full | 72.11 | 79.65 | 81.58 | 77.61 | 59.90 | 71.27 | 69.60 | 69.54 | 44.56 | 4.83 |
| 3 | D-LLM | 68.99 | 76.88 | 77.33 | 76.09 | 55.29 | 71.19 | 65.20 | 66.81 | 43.49 | 7.24 |
| | QTALE | 69.96 | 77.20 | 78.98 | 77.57 | 61.52 | 75.06 | 71.80 | **69.65** | **45.11** | **5.29** |

component should be normalized within a bounded range, and it should involve only a minimal number of parameters to allow practical adjustment. To this end, we apply softmax to the D-LLM router output, converting the class scores into probabilities within $[0, 1]$ that sum to 1. Since all routers produce probabilities under the same bounded distribution, a single global threshold $\theta$ can be shared across layers. Thus, the execution ratio of the entire model can be controlled in a lightweight, training-free manner with just one parameter $\theta$. Under this mechanism, a layer is executed if the probability for the execute class is greater than or equal to $\theta$, which can be expressed as:

$$b_l = \begin{cases} [1, 0], & \text{if } p_{l,0} \geq \theta \\ [0, 1], & \text{if } p_{l,0} < \theta \end{cases} \quad \text{where} \quad p_l = \text{softmax}(g_l(x_l)) \tag{9}$$

If $\theta = 0.5$, Eq. 9 becomes equivalent to the $\arg\max$-based decision rule in Eq. 2, since the class with the higher score is selected. Lowering $\theta$ below 0.5 increases the execution ratio by reducing the required probability for execution, whereas raising $\theta$ above 0.5 decreases the execution ratio by increasing this requirement. To adjust the execution threshold, we adopt a simple two-phase grid search strategy with a small calibration dataset. Since the objective is to

reintroduce redundancy, the threshold is searched within the range $(0, 0.5)$. In the coarse-grained phase, we sweep across the full target range using a large step size to quickly identify a promising region. In the fine-grained phase, we refine the search within a narrower window around the best coarse-phase candidate, employing a small step size to precisely determine the optimal threshold.

## 4. Experiments

### 4.1. Experimental Setup

**Models and Datasets.** We evaluate the proposed QTALE on three open-source LLMs: LLaMA2-7B, LLaMA3.1-8B, and LLaMA3.2-3B. For evaluation, we report zero-shot accuracy on the CommonsenseQA (CSQA) benchmark suite (Talmor et al., 2019), which includes PIQA, BoolQ, SIQA, ARCe, ARCc, Winogrande (Winogr.), and OBQA (Bisk et al., 2020; Clark et al., 2019; Sap et al., 2019; Clark et al., 2018; Sakaguchi et al., 2021; Mihaylov et al., 2018). We also evaluate zero-shot accuracy on the MMLU dataset (Hendrycks et al., 2021) and measure perplexity (PPL) on the Stanford-Alpaca dataset (Alpaca) (Taori et al., 2023) and SAMSum (Gliwa et al., 2019).

*Table 2.* Accuracy and PPL comparison on LLaMA3.2-3B. Accuracy is reported on CSQA and MMLU, while PPL is reported on Alpaca and Samsum. (Avg.: average)

| Bits | Layer Execution | CSQA | | | | | | | | MMLU (↑) | Alpaca (↓) | Samsum (↓) |
|---|---|---|---|---|---|---|---|---|---|---|---|---|
| | | PIQA | BoolQ | SIQA | ARCe | ARCc | Winogr. | OBQA | Avg. (↑) | | | |
| | Full | 84.98 | 87.58 | 77.94 | 82.41 | 69.54 | 81.69 | 79.40 | 80.51 | 53.91 | 3.62 | 4.02 |
| 16 | D-LLM | 83.90 | 62.92 | 78.30 | 82.28 | 69.03 | 78.93 | 78.00 | 76.19 | 54.02 | 4.95 | 4.96 |
| | QTALE | 84.66 | 84.09 | 77.38 | 83.25 | 69.62 | 79.79 | 79.00 | **79.68** | **54.64** | **4.91** | **4.79** |
| | Full | 81.12 | 86.36 | 75.59 | 80.09 | 65.27 | 76.80 | 77.60 | 77.55 | 47.95 | 3.71 | 4.15 |
| 4 | D-LLM | 82.37 | 62.37 | 76.56 | 80.35 | 66.21 | 75.06 | 74.80 | 73.96 | 51.62 | 5.22 | 5.20 |
| | QTALE | 82.97 | 85.90 | 76.36 | 81.31 | 67.49 | 77.66 | 77.20 | **78.41** | **52.12** | **4.26** | **4.89** |
| | Full | 64.64 | 79.44 | 66.38 | 67.55 | 51.96 | 65.90 | 66.00 | 65.98 | 36.79 | 5.20 | 5.31 |
| 3 | D-LLM | 64.36 | 62.56 | 57.32 | 68.43 | 47.70 | 65.90 | 51.60 | 59.70 | 40.57 | 7.33 | 6.07 |
| | QTALE | 71.06 | 62.74 | 61.72 | 69.23 | 48.81 | 66.46 | 59.60 | **62.82** | **42.09** | **5.54** | **5.64** |

**Baselines.** We compare the proposed QTALE against three baselines: the widely adopted PTQ method AWQ (Lin et al., 2024), the prior token-adaptive layer execution method D-LLM (Jiang et al., 2024), and their naive integration, evaluating them in terms of accuracy/PPL, model size (memory overhead), and FLOPs.

**Implementation Details.** In the experiments, all quantization is performed using the AWQ algorithm with a group size of 128 (Lin et al., 2024). We evaluate both 4-bit and 3-bit integer quantization settings. For token-adaptive layer execution, the fine-tuning configurations, including learning rate, number of training epochs, and other hyperparameters, for both the proposed QTALE and D-LLM follow the implementation details reported in the original D-LLM paper (Jiang et al., 2024).

### 4.2. Accuracy/PPL Evaluation

Table 1 and Table 2 present the accuracy and PPL results of the baseline methods and the proposed QTALE. The full-layer execution model refers to the fine-tuned LLMs on downstream tasks without applying token-adaptive layer execution. Across all benchmarks, token-adaptive layer execution models trained with the D-LLM approach and the proposed QTALE with $\mathcal{L}_{entropy}$ for quantization-robust training achieve comparable accuracy and PPL before quantization. However, when combined with quantization, the D-LLM approach suffers from noticeable drops in accuracy and PPL compared to the quantized full-layer execution models. In contrast, QTALE maintains performance close to that of the quantized full models. For example, on the CSQA benchmark with LLaMA2-7B, the accuracy of the 3-bit quantized full-layer execution model is 72.22%, while the 3-bit D-LLM model drops to 70.57%. With the proposed QTALE, the accuracy is recovered to 72.79%. A similar trend is even more pronounced in the LLaMA3.2-3B model. Under 4-bit quantization, QTALE achieves 78.41% accuracy on the CSQA benchmark, whereas the quantized

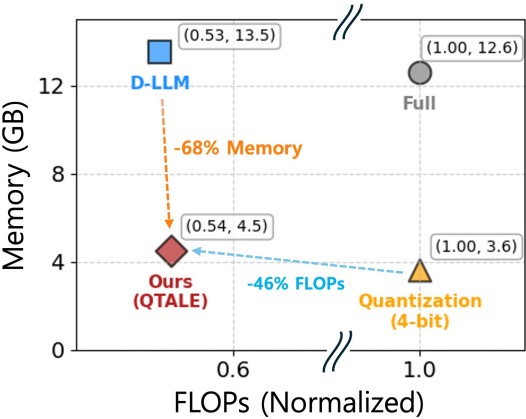

*Figure 6.* Efficiency trade-off between normalized FLOPs and memory footprint (GB) for LLaMA2-7B on the CSQA dataset.

full-layer execution baseline shows 77.55% accuracy, and D-LLM experiences a significant accuracy drop to 73.96%.

These results demonstrate that QTALE effectively restores the redundancy needed for robust quantization, enabling token-adaptive execution to be seamlessly integrated with low-bit quantization.

### 4.3. Efficiency Evaluation

**FLOPs and Memeory Usage.** Figure 6 presents the efficiency evaluation results in terms of model size (memory overhead) and FLOPs. With token-adaptive layer execution alone, the model size remains unchanged at 13.5 GB for LLaMA2-7B, making deployment on memory-constrained devices challenging. In contrast, when combined with 4-bit quantization, the model size is reduced to below 4.5 GB. With the proposed execution ratio adjustment mechanism, the execution ratio does not drastically increase on CSQA benchmarks, since these tasks can recover accuracy with only a slight increase in redundancy. Overall, these results demonstrate that the proposed approach enables dynamic

*Table 3.* Inference latency for processing 256 samples from each benchmark on an NVIDIA A6000 GPU (batch size 4; numbers in parentheses indicate speedup over the 16-bit full-model baseline).

| Layer Execute | PIQA | BoolQ | SIQA | CSQA ARCe | ARCc | Winogr. | OBQA | Avg. |
|---|---|---|---|---|---|---|---|---|
| **16-bit** | | | | | | | | |
| Full | 21.4 s (1.00x) | 27.4 s (1.00x) | 18.6 s (1.00x) | 27.9 s (1.00x) | 30.5 s (1.00x) | 11.1 s (1.00x) | 23.1 s (1.00x) | 22.8 s (1.00x) |
| D-LLM | 15.8 s (1.36x) | 20.1 s (1.37x) | 14.7 s (1.26x) | 21.8 s (1.28x) | 23.6 s (1.29x) | 9.3 s (1.20x) | 19.1 s (1.21x) | 17.8 s (1.28x) |
| QTALE | 16.3 s (1.31x) | 20.2 s (1.36x) | 14.6 s (1.28x) | 21.8 s (1.28x) | 24.5 s (1.25x) | 8.9 s (1.25x) | 18.5 s (1.25x) | 17.8 s (1.28x) |
| **4-bit** | | | | | | | | |
| Full | 22.2 s (0.96x) | 28.6 s (0.96x) | 19.5 s (0.95x) | 28.9 s (0.96x) | 30.6 s (1.00x) | 11.2 s (0.99x) | 23.8 s (0.97x) | 23.5 s (0.97x) |
| D-LLM | 17.1 s (1.25x) | 20.5 s (1.33x) | 15.0 s (1.24x) | 21.2 s (1.31x) | 23.5 s (1.30x) | 8.8 s (1.26x) | 18.5 s (1.25x) | 17.8 s (1.28x) |
| QTALE | 17.0 s (1.26x) | 21.1 s (1.30x) | 14.7 s (1.27x) | 21.6 s (1.29x) | 26.2 s (1.16x) | 8.6 s (1.29x) | 18.1 s (1.28x) | 18.2 s (1.26x) |

*Table 4.* Ablation study on the impact of the proposed quantization-robust training with $\mathcal{L}_{entropy}$ and execution ratio adjustment with $\theta$. Results are reported in terms of accuracy (Acc.), PPL, and $R_{avg}$ (average layer execution ratio) for token-adaptive layer execution models under 4-bit quantization.

| $\mathcal{L}_{entropy}$ | $\theta$ | CSQA Acc. | $R_{avg}$ | MMLU Acc. | $R_{avg}$ | Alpaca PPL | $R_{avg}$ |
|---|---|---|---|---|---|---|---|
| **LLaMA2-7B** | | | | | | | |
| x | 0.5 | 77.32 | 0.53 | 50.47 | 0.55 | 4.43 | 0.61 |
| x | adjust | 78.48 | 0.56 | 50.62 | 0.57 | 3.74 | 0.77 |
| o | 0.5 | 78.86 | 0.53 | 51.24 | 0.56 | 4.35 | 0.63 |
| o | adjust | 79.18 | 0.54 | 51.24 | 0.59 | 3.74 | 0.81 |
| **LLaMA3.1-8B** | | | | | | | |
| x | 0.5 | 77.68 | 0.54 | 55.36 | 0.55 | 5.47 | 0.60 |
| x | adjust | 78.14 | 0.59 | 55.36 | 0.56 | 4.30 | 0.76 |
| o | 0.5 | 78.86 | 0.53 | 55.80 | 0.55 | 5.01 | 0.63 |
| o | adjust | 79.17 | 0.54 | 55.86 | 0.55 | 4.11 | 0.80 |

adjustment of the execution ratio to balance efficiency and accuracy requirements. Additional results for other models and datasets are provided in the Appendix B.1, where they exhibit similar efficiency trends.

**Speedup.** To measure the actual speedup achievable with token-adaptive layer execution and quantization, we evaluate inference latency on the CSQA dataset. For each experiment, we randomly sampled 256 examples and ran inference on a single NVIDIA A6000 GPU (48 GB VRAM) with a batch size of 4. As shown in Table 3, the speedup achieved by D-LLM and QTALE is comparable. Token-adaptive layer execution yields an average speedup of 1.28×. However, there is no additional speedup when applying quantization, and QTALE shows a slight slowdown due to its increased execution ratio. Nevertheless, as shown in Figure 6, quantization provides a substantial reduction in memory usage, which is important in deployment scenarios with limited VRAM. For example, on a consumer-level GPU such as the NVIDIA RTX 5070 with only 12 GB of VRAM, the 4-bit quantized model runs successfully, whereas the 16-bit model triggers an out-of-memory error.

## 4.4. Additional Analysis

**Impact of Key Components.** The ablation study evaluates the impact of the two key components of QTALE: (1) quantization-robust training with $\mathcal{L}_{entropy}$ and (2) execution ratio adjustment with $\theta$. As shown in Table 4, both components play essential roles in recovering accuracy/PPL after quantization. The quantization-robust training with $\mathcal{L}_{entropy}$ increases the entropy of router logits, ensuring that the gap between execute and bypass logits remains small. As a result, it stabilizes path diversity during training but does not alter the execution ratio after fine-tuning. In contrast, the execution ratio adjustment with $\theta$ directly controls the number of executed layers. As discussed in the previous section, the amount of adjustment required to recover accuracy and PPL varies across models and benchmarks, depending on the level of redundancy needed for robustness.

**Compatibility with Other PTQ Techniques.** To demonstrate the effectiveness of QTALE when combined with other PTQ methods beyond AWQ, we evaluated its performance with the MagR+GPTQ quantization scheme (Zhang et al., 2024; Frantar et al., 2023). Table 5 presents the results when QTALE is paired with MagR. QTALE consistently outperforms D-LLM under both 4-bit and 3-bit quantization settings. These results confirm that QTALE presents the quantization-robustness regardless of PTQ methods and provides superior performance stability under quantization noise.

**Compatibility with Other Compression Techniques.** The proposed training scheme enhances quantization robustness by exposing the model to diverse execution paths during fine-tuning, which makes it generally resilient to inference-time perturbations that may alter the selected routing path. To assess its effectiveness when combined with other compression methods such as pruning, we apply 50% unstructured sparsity to a QTALE-trained model using Wanda (Sun et al., 2024) for weight pruning. Table 6 reports the corresponding performance. While D-LLM exhibits a substantial performance drop under 50% sparsity, QTALE compensates for the loss of weights and achieves performance comparable

*Table 5.* Accuracy and PPL of LLaMA3.2-3B under MagR+GPTQ quantization. (Avg.: average)

| Bits | Layer Execution | CSQA Avg. ($\uparrow$) | MMLU ($\uparrow$) | Alpaca ($\downarrow$) |
|---|---|---|---|---|
| 16 | Full | 80.51 | 53.91 | 3.62 |
| | D-LLM | 76.19 | 54.02 | 4.95 |
| | QTALE | **79.68** | **54.64** | **4.79** |
| 4 | Full | 76.61 | 47.83 | 4.05 |
| | D-LLM | 73.31 | 48.05 | 5.84 |
| | QTALE | **77.62** | **49.38** | **4.73** |
| 3 | Full | 65.17 | 36.34 | 5.10 |
| | D-LLM | 65.51 | 39.96 | 7.29 |
| | QTALE | **67.76** | **40.04** | **6.12** |

*Table 6.* Accuracy and PPL comparison on LLaMA3.2-3B for dense and pruned models. Accuracy is reported on CSQA and MMLU, while PPL is reported on Alpaca. (Avg.: average)

| Sparsity | Layer Execution | CSQA Avg. ($\uparrow$) | MMLU ($\uparrow$) | Alpaca ($\downarrow$) |
|---|---|---|---|---|
| 0% | Full | 80.51 | 53.91 | 3.62 |
| | D-LLM | 76.19 | 54.02 | 4.95 |
| | QTALE | **79.68** | **54.64** | **4.79** |
| 50% | Full | 68.35 | 40.46 | 5.76 |
| | D-LLM | 63.56 | **43.12** | 8.48 |
| | QTALE | **68.34** | 39.89 | **6.14** |

to the full-layer execution model. These results demonstrate that QTALE has strong potential to be extended to various post-training compression techniques.

## 5. Conclusion

To address the challenge of integrating token-adaptive layer execution with quantization for efficient LLM inference, this paper proposes QTALE, a novel framework that enables seamless integration of token-adaptive execution with quantization without sacrificing accuracy. QTALE introduces two key components: quantization-robust training with entropy regularization, which preserves training-path diversity, and inference-time execution ratio adjustment, which reintroduces redundancy when needed for robustness. Experimental results demonstrate that QTALE preserves accuracy after integrating token-adaptive execution with quantization, maintaining the gap to quantization-only models within 0.5% on CommonsenseQA, while simultaneously reducing both FLOPs and memory footprint. In summary, QTALE provides a practical and unified solution for efficient LLM deployment, effectively bridging the complementary benefits of token-adaptive execution and quantization.

## Acknowledgment

This work was supported in part by Institute of Information & communications Technology Planning & Evaluation (IITP) grant funded by the Korea government (MSIT) (No.RS-2025-02273157: Development of Low Power Training/Inference Technologies based on AI Semiconductors, RS-2026-25507427: Development of Efficient Architectures and Training Techniques for High-Performance Lightweight AI Models, RS-2025-10692981: AI Semiconductor Innovation Research Center (Sungkyunkwan University)), AI Computing Infrastructure Enhancement (GPU Rental Support) User Support Program funded by MSIT (No.RQT-25-090058), and Korea Institute for Advancement of Technology(KIAT) grant funded by the Korea Government (MOTIE) (P0023704, HRD Program for Industrial Innovation). (Corresponding Author: Yulhwa Kim).

## Impact Statement

This paper presents work whose goal is to advance the field of machine learning. There are many potential societal consequences of our work, none of which we feel must be specifically highlighted here.

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

# A. QTALE Details

## A.1. Experimental Settings

**Training Datasets.** Consistent with previous work D-LLM (Jiang et al., 2024), we use the official training split of each downstream benchmark—including CommonsenseQA tasks (PIQA, BoolQ, SIQA, ARCe, ARCc, Winogrande, and OBQA), MMLU, and Alpaca—for the fine-tuning stage of QTALE. That is, for each task, the model is adapted using the benchmark's official training set. For example, models fine-tuned on the Alpaca training set are evaluated on the Alpaca test set.

**Training Hyperparameter.** We largely follow the hyperparameter configurations of D-LLM. The learning rate is set to 0.009, $\lambda_1$ is set to 0.1 for all benchmarks except for the Alpaca fine-tuning case, where we use $\lambda_1 = 5$. We fine-tune for 10 epochs, but for the MMLU benchmark, we reduced the number of epochs from 10 to 3 due to its significantly larger training set compared to the other tasks. We consistently applied an entropy weight ($\lambda_2$) of 0.01 across all evaluated datasets. Regarding the batch size, LLaMA2-7B and LLaMA3.1-8B use various batch sizes as listed in Table 7, while for LLaMA3.2-3B we fixed the batch size to 4 across all training experiments for consistency.

*Table 7.* Batch sizes used for fine-tuning across different model configurations.

| Model | PIQA | BoolQ | SIQA | ARCe | ARCc | Winogr. | OBQA | MMLU | Alpaca |
|---|---|---|---|---|---|---|---|---|---|
| LLaMA2-7B | 4 | 4 | 8 | 1 | 1 | 4 | 2 | 7 | 7 |
| LLaMA3.1-8B | 4 | 4 | 7 | 6 | 6 | 4 | 6 | 6 | 7 |
| LLaMA3.2-3B | 4 | 4 | 4 | 4 | 4 | 4 | 4 | 4 | 4 |

**Threshold Calibration.** To calibrate the threshold, we use a calibration set of 300 samples and apply a simple grid-search procedure with a two-stage coarse-to-fine strategy. We first conduct a coarse search with a step size of 0.05 to identify the approximate optimal region, then perform a fine-grained search with a step size of 0.01 around the selected candidate.

## A.2. Sensitivity to Hyperparameter

**Sensitivity to the entropy regularization weight $\lambda_2$.** During fine-tuning, the other loss terms (e.g., execution-ratio loss) continue to decrease as training progresses, whereas the entropy loss saturates relatively quickly. Therefore, an overly large $\lambda_2$ suppresses the impact of these other losses. For this reason, we set $\lambda_2 = 0.1\lambda_1$. To analyze sensitivity, we scan $\lambda_2$ over a wide range from 0.005 to 0.5 on ARCc benchmark with LLaMA3.2-3B and reported the results in Figure 7. We observe:

- When $\lambda_2$ is near the default value $\lambda_2 = 0.1\lambda_1 = 0.01$, the accuracy remains stable with minimal variation.

- When $\lambda_2$ becomes substantially larger, the accuracy begins to fluctuate and slightly decreases.

Overall, QTALE is insensitive to $\lambda_2$ within a reasonable neighborhood around the chosen value, and performance remains stable.

**Sensitivity to the inference threshold $\theta$** We also evaluate LLaMA3.2-3B across a wide range of inference thresholds $\theta$,

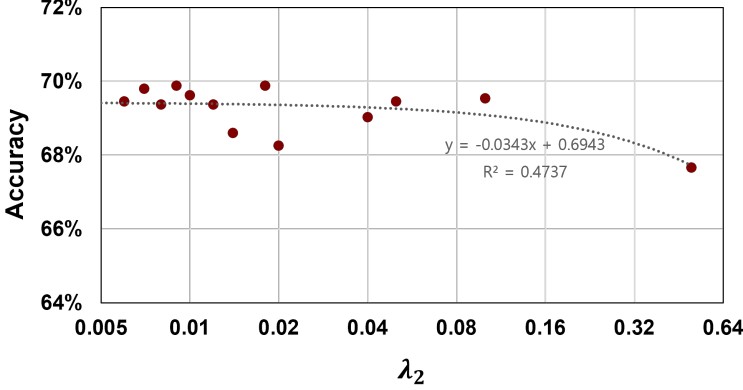

*Figure 7.* Accuracy of LLaMA3.2-3B on the ARCc benchmark after fine-tuning with different $\lambda_2$ settings. The dashed line indicates the trend line.

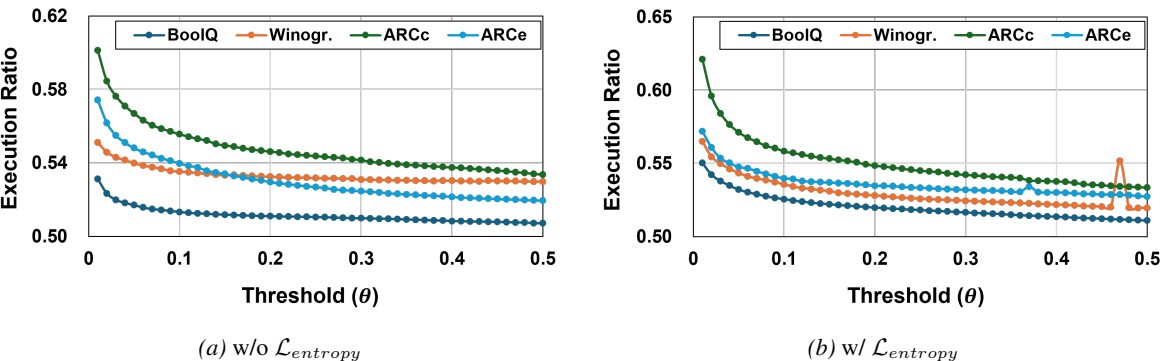

*(a)* w/o $\mathcal{L}_{entropy}$                    *(b)* w/ $\mathcal{L}_{entropy}$

*Figure 8.* Execution Ratio–Threshold curves for four representative CSQA tasks (ARCc, ARCe, BoolQ, and Winogrande).

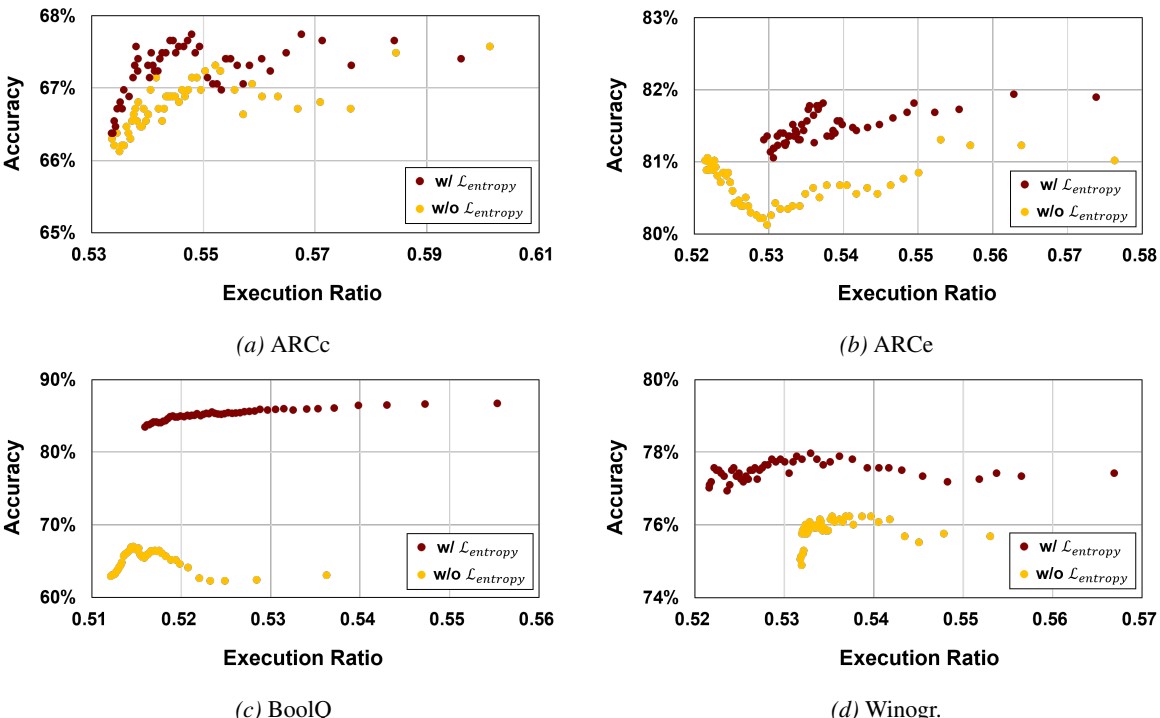

*(a)* ARCc                    *(b)* ARCe

*(c)* BoolQ                    *(d)* Winogr.

*Figure 9.* Accuracy–Execution Ratio curves for four representative CSQA tasks (ARCC, ARCE, BoolQ, and Winogr.). We compare the baseline without $\mathcal{L}_{entropy}$ (D-LLM) against the proposed method with $\mathcal{L}_{entropy}$ (QTALE) on LLaMA3.2-3B.

with results shown in Figures 8 and 9. Our observations are:

- As $\theta$ decreases, the execution ratio generally increases, although the magnitude differs across tasks (Figure 8).

- For models trained with entropy regularization, accuracy improves as the execution ratio increases and eventually saturates (Figure 9).

- For models trained without entropy regularization (i.e., the D-LLM training setup), higher execution ratios do not consistently improve accuracy and can even degrade performance indicating poor adaptability to diverse execution paths, as discussed in Section 3.1 (Figure 9).

This behavior demonstrates that QTALE induces a monotonic and predictable relationship between the threshold and accuracy, enabling a simple calibration strategy (e.g., grid search) to reliably select an appropriate threshold.

## A.3. Additional Analysis of Integrating Token-Adaptive Layer Execution and Quantization

The primary reason accuracy degrades when combining token-adaptive execution with quantization is that quantization significantly disrupts the learned routing patterns. Token-adaptive models (e.g., D-LLM) assume a stable execution path after fine-tuning. However, low-bit quantization perturbs hidden states and router logits, causing substantial path drift.

As shown in Table 8, under 3-bit quantization, 39.75% of tokens in SIQA and 24.68% of tokens in OBQA switch their execution path compared to the FP16 model. This drift often pushes tokens toward pruned, lower-capacity paths and alters routing boundaries, directly harming accuracy. QTALE mitigates this effect by employing an entropy-based objective that restores path redundancy and produces smoother, more quantization-robust routing behavior. Consequently, QTALE maintains accuracy close to quantized full models, unlike prior token-adaptive approaches.

*Table 8.* Token path change statistics under 3-bit quantization.

| Dataset | Bits | Total Tokens | Path Changed | Path Change Rate |
|---------|------|--------------|--------------|------------------|
| SIQA    | 3    | 194,518      | 77,326       | 39.75%           |
| OBQA    |      | 43,768       | 10,800       | 24.68%           |

## A.4. Additional Analysis of the Impact of Entropy Regularization across Layers

*Table 9.* Gumbel noise–induced execution decision flipping ratio of Llama3.1-8B

| Dataset  | Method | Epoch 6 | Epoch 7 | Epoch 8 | Epoch 9 |
|----------|--------|---------|---------|---------|---------|
| ARCE     | Ours   | **0.0459** | **0.0365** | **0.0425** | **0.0429** |
|          | D-LLM  | 0.0246  | 0.0338  | 0.0330  | 0.0327  |
| BoolQ    | Ours   | **0.0288** | **0.0331** | **0.0326** | **0.0321** |
|          | D-LLM  | 0.0153  | 0.0149  | 0.0136  | 0.0135  |
| PIQA     | Ours   | **0.0236** | **0.0261** | **0.0238** | **0.0239** |
|          | D-LLM  | 0.0181  | 0.0158  | 0.0162  | 0.0145  |
| Winogra. | Ours   | **0.0307** | **0.0325** | **0.0337** | **0.0342** |
|          | D-LLM  | 0.0178  | 0.0164  | 0.0137  | 0.0143  |

To clearly validate the impact of entropy regularization across the network, we analyze the average flipping ratio of router decisions for layers 16–31 at later training stages (epochs 6–9) for both D-LLM and our method, as shown in the Table 9.

We focus on these layers and training stages because most layers are consistently executed in early layers or during the initial phase of training (Figure 2). Entropy regularization is designed to maintain router entropy in a regime that enables decision flipping during training by regulating the execution and bypass logits within a range where Gumbel noise can flip the decision. This prevents premature pruning of unexecuted paths. Therefore, its effect is best examined in deeper layers and later training stages, where path-pruning behavior is more likely to occur.

As shown in Table 9, our method maintains consistently higher flipping ratios across layers 16–31 during epochs 6–9. It indicates that router decisions remain active rather than collapsing to a single outcome. This suggests that our approach mitigates the path pruning issue observed in D-LLM through entropy regularization, thereby preserving training-path diversity.

### A.5. Additional Analysis of How Execution Ratio Adjustment Compensates for Quantization Error

We can define the total error ($e_{total}$) as the sum of the error introduced by layer skipping ($e_{skip}$) and quantization ($e_{quant}$) as follows:

$$e_{total} = e_{skip} + e_{quant} \tag{10}$$

Quantization increases $e_{quant}$, while layer skipping introduces $e_{skip}$. As quantization becomes more aggressive, the combined error increases. Execution ratio adjustment compensates for this by increasing the number of executed layers, thereby reducing $e_{skip}$ and balancing the increase in $e_{quant}$.

To empirically validate this decomposition, we measure the cosine similarity between the dense FP16 model output and the outputs of models with quantization and/or layer skipping. The experiment is conducted under 3-bit quantization using 150 samples from the ARC-e dataset, where we explicitly control layer skipping by forcing the model to skip the 23rd layer.

As shown in Table 10, under quantization, executing the layer yields higher similarity to the dense model (0.8605) than skipping the layer (0.8219). This indicates that skipping introduces additional deviation, while executing the layer reduces this error.

This observation is consistent with our formulation: skipping increases $e_{skip}$, whereas executing reduces it, thereby lowering $e_{total}$. As a result, increasing the execution ratio effectively compensates for quantization-induced error, leading to outputs that better align with the dense model.

*Table 10.* Cosine similarity (Cos.) between quantized/layer-skipped model outputs and the dense FP16 output.

| Quant. | Skip Layer | Cos. |
|:------:|:----------:|:------:|
| x | o | 0.9610 |
| o | o | 0.8219 |
| o | x | 0.8605 |

# B. Additional Evaluation

## B.1. Detailed Efficiency Evaluation on Additional Models and Datasets

While Figure 6 presents efficiency evaluation results in terms of model size (memory overhead) and FLOPs using LLaMA2-7B on the CSQA dataset as a representative example, Table 11 reports detailed results for other models (e.g., LLaMA3.1-8B) and datasets (e.g., MMLU and Alpaca), exhibiting consistent efficiency trends. Applying token-adaptive layer execution alone does not reduce the model size of LLaMA3.1-8B, which remains at 16.0 GB, posing challenges for deployment on memory-constrained devices. In contrast, when combined with 4-bit quantization, our approach reduces the model size to below 6.3 GB.

In terms of computational efficiency, the required execution ratio varies with task difficulty. For example, CSQA and MMLU recover accuracy with only a marginal increase in redundancy, whereas the Alpaca benchmark requires a higher execution ratio to preserve perplexity (PPL) for LLaMA3.1-8B. Overall, these results demonstrate that the proposed approach enables dynamic adjustment of the execution ratio, effectively balancing efficiency and accuracy across different benchmarks and models.

*Table 11.* Model size and FLOPs required for single-token processing with LLaMA2-7B and LLaMA3.1-8B. Numbers in parentheses denote FLOPs relative to full-model execution.

| | | LLaMA2-7B | | | | LLaMA3.1-8B | | | |
| | | Model | FLOPs | | | Model | FLOPs | | |
| Bits | Layer Execution | (GB) | CSQA | MMLU | Alpaca | (GB) | CSQA | MMLU | Alpaca |
|---|---|---|---|---|---|---|---|---|---|
| | Full | 12.6 | 13.0 (1.00x) | 13.0 (1.00x) | 13.0 (1.00x) | 15.0 | 15.6 (1.00x) | 15.6 (1.00x) | 15.6 (1.00x) |
| 16 | D-LLM | 13.5 | 6.92 (0.54x) | 7.18 (0.55x) | 7.76 (0.60x) | 16.0 | 8.39 (0.54x) | 8.53 (0.55x) | 9.30 (0.60x) |
| | QTALE | | 6.81 (0.52x) | 7.27 (0.56x) | 8.03 (0.62x) | | 8.24 (0.53x) | 8.55 (0.55x) | 9.65 (0.62x) |
| | Full | 3.6 | 13.0 (1.00x) | 13.0 (1.00x) | 13.0 (1.00x) | 5.3 | 15.6 (1.00x) | 15.6 (1.00x) | 15.6 (1.00x) |
| 4 | D-LLM | 4.5 | 6.85 (0.53x) | 7.62 (0.59x) | 7.86 (0.61x) | 6.3 | 8.46 (0.54x) | 8.53 (0.55x) | 9.30 (0.60x) |
| | QTALE | | 6.97 (0.54x) | 7.69 (0.59x) | 10.47 (0.81x) | | 8.44 (0.54x) | 8.57 (0.55x) | 12.53 (0.80x) |
| | Full | 2.8 | 13.0 (1.00x) | 13.0 (1.00x) | 13.0 (1.00x) | 4.5 | 15.6 (1.00x) | 15.6 (1.00x) | 15.6 (1.00x) |
| 3 | D-LLM | 3.8 | 7.21 (0.56x) | 7.46 (0.58x) | 8.24 (0.64x) | 5.5 | 8.44 (0.54x) | 8.59 (0.55x) | 9.49 (0.61x) |
| | QTALE | | 7.13 (0.55x) | 7.63 (0.59x) | 11.01 (0.85x) | | 8.54 (0.55x) | 8.66 (0.56x) | 12.91 (0.83x) |

## B.2. Practical Latency Analysis under Large Batch Sizes

*Table 12.* Inference latency (seconds) and relative speedup across batch sizes on NVIDIA A6000.

| Bits | Method | Batch 8 | Batch 16 | Batch 32 |
|---|---|---|---|---|
| | Full | 22.9 (1.00x) | 24.0 (1.00x) | 25.8 (1.00x) |
| 16 | D-LLM | 16.7 (1.37x) | 16.9 (1.42x) | 17.9 (1.44x) |
| | QTALE | 16.7 (1.37x) | 17.0 (1.41x) | 18.2 (1.42x) |
| | Full | 24.6 (0.93x) | 26.2 (0.92x) | 29.3 (0.88x) |
| 4 | D-LLM | 17.5 (1.31x) | 18.3 (1.31x) | 19.6 (1.31x) |
| | QTALE | 18.0 (1.27x) | 19.1 (1.25x) | 19.9 (1.29x) |

To further analyze the practical speedup of QTALE, we evaluate inference latency under larger batch sizes (8, 16, and 32) on a single NVIDIA A6000 GPU (48 GB VRAM), following the same evaluation protocol as Table 3. Reported values denote average latency over all CSQA tasks.

As shown in Table 12, QTALE consistently achieves lower latency than the full-precision model across all batch sizes, demonstrating that the proposed dynamic execution strategy maintains stable practical speedup under batched inference. Although QTALE exhibits slightly lower speedup than D-LLM, this tradeoff arises from the increased execution ratio used to recover robustness under quantization.

Latency increases with batch size primarily because static batching is used during evaluation, causing shorter sequences to wait for longer sequences through padding. To ensure a fair comparison, padding tokens are fully processed in the full model, while dynamically skipped in D-LLM and QTALE according to routing decisions.

To reduce fragmented execution overhead, active tokens selected by the router are gathered into dense inputs before each layer execution. This avoids inefficient per-token computation and preserves batched execution over active tokens, making the latency improvement robust across different batch sizes. We further implement custom Triton kernels for dynamic sub-layer skipping, including fused normalization and activation kernels together with a custom routing-decision kernel. Although the achieved latency speedup does not fully match the theoretical FLOPs reduction due to remaining kernel-launch overhead and control-flow inefficiencies, the current implementation consistently demonstrates practical latency improvements over full-model execution. Further optimization of the router block and more aggressive kernel fusion remain important future directions for improving hardware efficiency.

## B.3. Comparison with other Efficiency Technique

To compare QTALE against other efficiency methods, we additionally evaluate structured pruning combined with quantization. Since layer-wise pruning is known to provide a better speedup–accuracy trade-off than other structured pruning strategies for LLMs, we adopted SLEB, a state-of-the-art layer-wise pruning method, for this experiment (Song et al., 2024). The evaluation is conducted on LLaMA2-7B, and the results are shown in Table 13. In terms of speedup, 20% layer pruning achieves a similar acceleration to QTALE, yielding approximately $1.25\times$ over the baseline full-layer execution model. Compared to QTALE, structured pruning provides better memory efficiency because redundant layers are removed entirely. However, the accuracy degradation under 20% pruning is substantially larger than that of QTALE (Note that accuracy of QTALE is reported on Table 1). Even when we consider a milder setting such as 10% layer pruning, QTALE consistently achieves better accuracy and perplexity. These results highlight that QTALE offers a more favorable accuracy–efficiency trade-off compared to structured pruning combined with quantization.

*Table 13.* Accuracy, PPL, and model size for SLEB-pruned LLaMA2-7B under 4-bit and 3-bit AWQ quantization. Avg. denotes the average CSQA accuracy.

| Sparsity | Model (GB) | Bits | PIQA | BoolQ | SIQA | ARCe | ARCc | Winogr. | OBQA | Avg. (↑) | MMLU (↑) | Alpaca (↓) |
|---|---|---|---|---|---|---|---|---|---|---|---|---|
| | 11.0 | 16 | 83.19 | 86.82 | 72.01 | 79.97 | 61.86 | 68.90 | 77.00 | 75.68 | 46.94 | 3.61 |
| 10% | 2.9 | 4 | 80.20 | 86.88 | 71.03 | 76.68 | 58.53 | 62.43 | 74.60 | 72.91 | 44.87 | 3.58 |
| | 2.2 | 3 | 71.16 | 77.15 | 64.48 | 64.73 | 45.73 | 56.75 | 55.20 | 62.17 | 34.71 | 4.45 |
| | 9.9 | 16 | 76.77 | 75.07 | 67.20 | 73.53 | 52.73 | 60.62 | 65.00 | 67.27 | 44.06 | 4.71 |
| 20% | 2.6 | 4 | 72.85 | 72.32 | 65.66 | 68.94 | 47.87 | 55.56 | 60.20 | 63.34 | 42.35 | 4.76 |
| | 1.9 | 3 | 53.97 | 48.55 | 49.08 | 44.02 | 30.38 | 54.38 | 25.20 | 43.65 | 31.56 | 7.54 |

## B.4. Mixed-Precision Quantization

To evaluate QTALE under more challenging precision settings, we consider a mixed-precision configuration that combines 2-bit and 4-bit weights. Specifically, we interleave 4-bit and 2-bit quantization in a layer-wise manner ($i$-th layer: 4-bit, $(i + 1)$-th layer: 2-bit). This mixed-precision setup introduces substantially higher quantization noise. We apply MagR+GPTQ for quantization (Zhang et al., 2024; Frantar et al., 2023) and evaluate the LLaMA2-7B model. As shown in Table 14, even under these demanding conditions, QTALE consistently provides stronger quantization robustness compared to D-LLM.

*Table 14.* Evaluation results of mixed 2&4-bit quantization on the LLaMA2-7B model.

| Layer Execution | PIQA | BoolQ | SIQA | ARCe | ARCc | Winogr. | OBQA | Avg. | MMLU | Alpaca |
|---|---|---|---|---|---|---|---|---|---|---|
| | | | | **Accuracy** | | | | | | |
| Full | 60.17 | 70.51 | 62.44 | 66.75 | 38.40 | 53.73 | 60.40 | 58.91 | 30.54 | 12.67 |
| D-LLM | 61.59 | 61.36 | 34.03 | 63.34 | 47.61 | 50.43 | 59.00 | 53.91 | **30.99** | 61.25 |
| QTALE | 61.04 | 63.90 | 50.15 | 65.74 | 49.32 | 51.07 | 62.20 | **57.63** | 27.05 | **35.20** |
| | | | | **Execution Ratio** | | | | | | |
| D-LLM | 0.5103 | 0.5100 | 0.5945 | 0.5274 | 0.6115 | 0.7781 | 0.5639 | 0.5851 | 0.5554 | 0.6578 |
| QTALE | 0.5131 | 0.5443 | 0.6099 | 0.5337 | 0.6376 | 0.8740 | 0.5815 | 0.6134 | 0.5827 | 0.8306 |
| | | | | **Threshold ($\theta$)** | | | | | | |
| QTALE | 0.05 | 0.45 | 0.49 | 0.41 | 0.38 | 0.41 | 0.45 | – | 0.20 | 0.05 |

## B.5. Other Optimal Threshold Searching Methods

Beyond the heuristic grid search used for threshold calibration in QTALE, we explored several optimization-based approaches for threshold tuning. We primarily examined two categories of methods: evolution strategies and Bayesian optimization. For the evolution-strategy approach, we adopted Natural Evolution Strategies (NES) (Wierstra et al., 2014), which iteratively updates the threshold distribution through gradient-based estimation. For Bayesian optimization, we tested both (i) a standard Gaussian Process–based Bayesian optimizer provided as a Python package (Nogueira, 2014–) (denoted as Bayes) and (ii) the Bayesian optimization algorithm implemented in the Optuna framework (Akiba et al., 2019), which leverages the Tree-structured Parzen Estimator (TPE). As shown in Table 15, these more advanced optimization methods are often able to discover thresholds that yield slightly better accuracy. However, when considering the overall calibration time, the simple grid search remains a competitive and practical choice.

*Table 15.* Performance comparison of different threshold calibration strategies.

| Method | PIQA | BoolQ | SIQA | CSQA ARCe | ARCc | Winogr. | OBQA | Avg. |
|---|---|---|---|---|---|---|---|---|
| **Accuracy** | | | | | | | | |
| grid | 76.20 | 82.86 | 85.90 | 81.31 | 67.49 | 77.66 | 77.20 | 78.37 |
| nes | 76.46 | 83.46 | 86.08 | 81.82 | 67.75 | 78.22 | 76.60 | 78.63 |
| bayes | 76.71 | 83.79 | 86.17 | 81.82 | 67.75 | 78.77 | 77.00 | 78.86 |
| optuna | 76.66 | 83.73 | 86.23 | 81.73 | 67.66 | 78.69 | 77.60 | 78.90 |
| **Execution Ratio** | | | | | | | | |
| grid | 0.5425 | 0.5244 | 0.5511 | 0.5505 | 0.5448 | 0.5258 | 0.5385 | 0.5396 |
| nes | 0.5177 | 0.5259 | 0.5387 | 0.5349 | 0.5488 | 0.5203 | 0.5326 | 0.5313 |
| bayes | 0.5198 | 0.5251 | 0.5483 | 0.5347 | 0.5441 | 0.5218 | 0.5381 | 0.5331 |
| optuna | 0.5195 | 0.5253 | 0.5452 | 0.5333 | 0.5444 | 0.5217 | 1.0000 | 0.7013 |
| **Best Threshold ($\theta$)** | | | | | | | | |
| grid | 0.0500 | 0.0300 | 0.0600 | 0.0400 | 0.2500 | 0.2500 | 0.1000 | – |
| nes | 0.0120 | 0.0967 | 0.3930 | 0.1965 | 0.1952 | 0.4542 | 0.0664 | – |
| bayes | 0.3835 | 0.1042 | 0.1460 | 0.2024 | 0.2675 | 0.4125 | 0.0760 | – |
| optuna | 0.3932 | 0.1017 | 0.1776 | 0.2442 | 0.2625 | 0.4140 | 0.0000 | – |
| **Calibration Time (s)** | | | | | | | | |
| grid | 803.83 | 621.55 | 707.16 | 1051.51 | 959.80 | 576.62 | 940.97 | 808.78 |
| nes | 2476.38 | 1928.49 | 2425.95 | 3304.38 | 3606.26 | 1620.04 | 2994.07 | 2622.22 |
| bayes | 832.21 | 2216.38 | 717.25 | 2624.23 | 999.65 | 1862.74 | 3108.58 | 1765.85 |
| optuna | 1206.63 | 1589.58 | 922.49 | 1888.10 | 1459.43 | 1374.25 | 2053.64 | 1499.16 |

## B.6. Detailed Execution Ratio and Threshold Results

In this section, we provide detailed results of the layer execution ratios and the corresponding thresholds ($\theta$) used across our experiment. Tables 16 and 17 provides execution ratios and thresholds ($\theta$) for LLaMA2-7B and LLaMA3.1-8B, which correspond to the results reported in Table 1. Tables 18 and 19 provides execution ratios and thresholds ($\theta$) for LLaMA3.2-3B, corresponding to the results reported in Tables 2 and 6, respectively.

*Table 16.* Execution Ratio and threshold results for LLaMA2-7B

| Bits | Layer Execution | CSQA | | | | | | | Avg. | MMLU | Alpaca |
| | | PIQA | BoolQ | SIQA | ARCe | ARCc | Winogr. | OBQA | | | |
| --- | --- | --- | --- | --- | --- | --- | --- | --- | --- | --- | --- |
| **Execution Ratio** | | | | | | | | | | | |
| 16 | D-LLM | 0.5088 | 0.5306 | 0.5087 | 0.5209 | 0.5943 | 0.5163 | 0.5591 | 0.5367 | 0.5546 | 0.5989 |
| | QTALE | 0.5069 | 0.5203 | 0.5319 | 0.5209 | 0.5531 | 0.5238 | 0.5222 | 0.5300 | 0.5611 | 0.6203 |
| 4 | D-LLM | 0.5128 | 0.5756 | 0.5888 | 0.5228 | 0.5361 | 0.5250 | 0.5613 | 0.5513 | 0.5884 | 0.6066 |
| | QTALE | 0.5284 | 0.5275 | 0.5322 | 0.5334 | 0.5747 | 0.5366 | 0.5353 | 0.5453 | 0.5941 | 0.8086 |
| 3 | D-LLM | 0.5141 | 0.5825 | 0.5875 | 0.5234 | 0.6002 | 0.5251 | 0.5628 | 0.5590 | 0.5763 | 0.6359 |
| | QTALE | 0.5544 | 0.5269 | 0.5319 | 0.5584 | 0.6099 | 0.5263 | 0.5441 | 0.5551 | 0.5886 | 0.8504 |
| **Threshold ($\theta$)** | | | | | | | | | | | |
| 16 | | 0.50 | 0.50 | 0.50 | 0.50 | 0.50 | 0.50 | 0.50 | – | 0.50 | 0.50 |
| 4 | QTALE | 0.25 | 0.30 | 0.50 | 0.23 | 0.12 | 0.22 | 0.15 | – | 0.05 | 0.05 |
| 3 | | 0.05 | 0.35 | 0.50 | 0.10 | 0.22 | 0.40 | 0.15 | – | 0.08 | 0.05 |

*Table 17.* Execution Ratio and threshold results for LLaMA3.1-8B

| Bits | Layer Execution | CSQA | | | | | | | Avg. | MMLU | Alpaca |
| | | PIQA | BoolQ | SIQA | ARCe | ARCc | Winogr. | OBQA | | | |
| --- | --- | --- | --- | --- | --- | --- | --- | --- | --- | --- | --- |
| **Execution Ratio** | | | | | | | | | | | |
| 16 | D-LLM | 0.5244 | 0.5241 | 0.5691 | 0.5341 | 0.5172 | 0.5272 | 0.5772 | 0.5402 | 0.5481 | 0.5972 |
| | QTALE | 0.5068 | 0.5070 | 0.5317 | 0.5307 | 0.5517 | 0.5340 | 0.5436 | 0.5319 | 0.5494 | 0.6199 |
| 4 | D-LLM | 0.5244 | 0.5241 | 0.5691 | 0.5341 | 0.5172 | 0.5272 | 0.5772 | 0.5402 | 0.5481 | 0.5972 |
| | QTALE | 0.5066 | 0.5077 | 0.5569 | 0.5671 | 0.5666 | 0.5334 | 0.5572 | 0.5432 | 0.5503 | 0.8046 |
| 3 | D-LLM | 0.5344 | 0.5206 | 0.5700 | 0.5353 | 0.5384 | 0.5256 | 0.5719 | 0.5435 | 0.5516 | 0.6094 |
| | QTALE | 0.5086 | 0.5059 | 0.5639 | 0.5869 | 0.5951 | 0.5381 | 0.5401 | 0.5494 | 0.5564 | 0.8294 |
| **Threshold ($\theta$)** | | | | | | | | | | | |
| 16 | | 0.50 | 0.50 | 0.50 | 0.50 | 0.50 | 0.50 | 0.50 | – | 0.50 | 0.50 |
| 4 | QTALE | 0.30 | 0.50 | 0.25 | 0.20 | 0.25 | 0.50 | 0.10 | – | 0.45 | 0.05 |
| 3 | | 0.10 | 0.50 | 0.22 | 0.05 | 0.08 | 0.50 | 0.46 | – | 0.32 | 0.05 |

*Table 18.* Execution ratio and threshold results for LLaMA3.2-3B

| Bits | Layer Execution | CSQA | | | | | | | Avg | MMLU | Alpaca | Samsum |
|---|---|---|---|---|---|---|---|---|---|---|---|---|
| | | PIQA | BoolQ | SIQA | ARCe | ARCc | Winogr. | OBQA | | | | |
| **Execution Ratio** | | | | | | | | | | | | |
| 16 | D-LLM | 0.5095 | 0.5074 | 0.5343 | 0.5197 | 0.5321 | 0.5285 | 0.5301 | 0.5278 | 0.5607 | 0.6132 | 0.5952 |
| | QTALE | 0.5349 | 0.5158 | 0.5097 | 0.5252 | 0.5336 | 0.5204 | 0.5316 | 0.5273 | 0.5469 | 0.6107 | 0.5947 |
| 4 | D-LLM | 0.5099 | 0.5069 | 0.5292 | 0.5196 | 0.5336 | 0.5297 | 0.5309 | 0.5275 | 0.5604 | 0.6199 | 0.5978 |
| | QTALE | 0.5425 | 0.5163 | 0.5511 | 0.5505 | 0.5448 | 0.5258 | 0.5385 | 0.5433 | 0.5767 | 0.7775 | 0.6832 |
| 3 | D-LLM | 0.5095 | 0.5090 | 0.5285 | 0.5263 | 0.5345 | 0.5317 | 0.5287 | 0.5285 | 0.5598 | 0.6382 | 0.6045 |
| | QTALE | 0.5459 | 0.5226 | 0.5472 | 0.5347 | 0.5450 | 0.5284 | 0.5441 | 0.5401 | 0.5528 | 0.7910 | 0.6902 |
| **Threshold ($\theta$)** | | | | | | | | | | | | |
| 16 | | 0.50 | 0.50 | 0.50 | 0.50 | 0.50 | 0.50 | 0.50 | – | 0.50 | 0.50 | 0.50 |
| 4 | QTALE | 0.05 | 0.03 | 0.06 | 0.04 | 0.25 | 0.25 | 0.10 | – | 0.26 | 0.05 | 0.05 |
| 3 | | 0.09 | 0.15 | 0.12 | 0.20 | 0.21 | 0.30 | 0.16 | – | 0.3 | 0.05 | 0.05 |

*Table 19.* Execution ratio and threshold results for LLaMA3.2-3B on dense and pruned models.

| Sparsity | Layer Execution | CSQA | | | | | | | Avg | MMLU | Alpaca |
|---|---|---|---|---|---|---|---|---|---|---|---|
| | | PIQA | BoolQ | SIQA | ARCe | ARCc | Winogr. | OBQA | | | |
| **Execution Ratio** | | | | | | | | | | | |
| 0% | D-LLM | 0.5095 | 0.5074 | 0.5343 | 0.5197 | 0.5321 | 0.5285 | 0.5301 | 0.5373 | 0.5607 | 0.6132 |
| | QTALE | 0.5158 | 0.5097 | 0.5349 | 0.5252 | 0.5336 | 0.5204 | 0.5316 | 0.5469 | 0.5469 | 0.6107 |
| 50% | D-LLM | 0.5115 | 0.5065 | 0.5349 | 0.5188 | 0.5276 | 0.5259 | 0.5298 | 0.5221 | 0.5544 | 0.5946 |
| | QTALE | 0.5196 | 0.5336 | 0.5576 | 0.5310 | 0.5813 | 0.5476 | 0.5382 | 0.5441 | 0.5489 | 0.7644 |
| **Threshold ($\theta$)** | | | | | | | | | | | |
| 0% | QTALE | 0.50 | 0.50 | 0.50 | 0.50 | 0.50 | 0.50 | 0.50 | – | 0.50 | 0.50 |
| 50% | | 0.36 | 0.05 | 0.04 | 0.29 | 0.03 | 0.05 | 0.17 | – | 0.17 | 0.05 |

## B.7. Evaluation on Complex Reasoning Tasks

We evaluate zero-shot accuracy of Llama3.2-3B and Llama3.1-8B on the GSM8K benchmark using input prompts for Chain-of-Thought (CoT) reasoning. We use the same hyperparameter settings as in our previous experiments (e.g., $\lambda_2 = 0.01$). While we use 300 samples for threshold calibration in other benchmarks, we use 64 samples for GSM8K. This is because CoT reasoning produces longer output sequences, and we adjust the calibration size to maintain a comparable calibration overhead across benchmarks.

As shown in Table 20, QTALE consistently achieves higher accuracy than D-LLM after quantization. For example, the accuracy of 4-bit D-LLM and QTALE is 16.98% and 23.28%, respectively, for Llama3.2-3B. These results demonstrate that our method remains effective on mathematical reasoning tasks.

## B.8. Evaluation on Instruction-Following Benchmark

With the LLaMA3.2-3B model fine-tuned on the Alpaca dataset, we evaluate the instruction-following capabilities of both D-LLM and QTALE using AlpacaEval (Li et al., 2023). GPT-4o mini is used as the evaluator. As shown in Figure 10, QTALE and D-LLM exhibit similar instruction-following performance under the 16-bit setting. However, when applying 4-bit AWQ quantization, QTALE significantly outperforms D-LLM, achieving a 70.46% win rate. We additionally provide qualitative comparisons of instruction-following outputs under the 4-bit AWQ configuration in Examples 1–5, which further highlight the quantization robustness of QTALE.

*Table 20.* GSM8K Accuracy (%).

| Bits | Method | Acc. (%) ↑ |
|---|---|---|
| | **LLaMA3.1-8B** | |
| 16 | Full | 45.26 |
| | D-LLM | 41.85 |
| | QTALE | **45.03** |
| 4 | Full | 35.33 |
| | D-LLM | 32.60 |
| | QTALE | **42.83** |
| | **LLaMA3.2-3B** | |
| 16 | Full | 30.50 |
| | D-LLM | 26.38 |
| | QTALE | **26.54** |
| 4 | Full | 20.39 |
| | D-LLM | 16.98 |
| | QTALE | **23.28** |

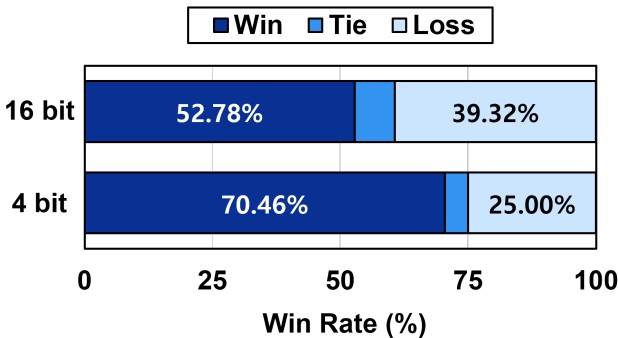

*Figure 10.* Win rates of QTALE over D-LLM on AlpacaEval, evaluated using GPT-4o mini. Results are reported for both full-precision (16-bit) and quantized (4-bit) weight settings using the LLaMA3.2-3B model.

---

**Example 1**

**Instruction:** What are some species of bears that are now extinct?
**Model Outputs:**

- **QTALE (4-bit quantized):** Unfortunately, many species of bears have become extinct. Some include the American bear, spectacled bear, Chinese giant panda, Himalayan bear, and polar bear.

- **D-LLM (4-bit quantized):** Answer Jonathan Crusher is a biologist at the University of Illinois. He has studied the evolution of the bear species for over 10 years. He has also been a consultant on the movie "The Bear King", released by Universal Studios in 2019.

**Instruction:** Why do a lot of scientists not believe in God or Satan?
**Model Outputs:**

- **QTALE (4-bit quantized):** Many scientists believe scientific explanations are sufficient and see no need for supernatural beings. Others argue religious beliefs lack empirical evidence.

- **D-LLM (4-bit quantized):** Why do you, who a toeh and forth between between belief-bel Bel or vice vice vice.

## Example 2

**Instruction:** Please tell me the history of Walt Disney.
**Model Outputs:**

- **QTALE (4-bit quantized):** I heard somewhere that it was started by two men, but I don't remember. He moved between studios and had to pay royalties before using some works.

- **D-LLM (4-bit quantized):** how they became an incredibly great leader leaders of the time period.

**Instruction:** How are carbon fibers used in buildings?
**Model Outputs:**

- **QTALE (4-bit quantized):** Carbon fibers reinforce structural materials, strengthen electronics, sports equipment, aerospace components, and medical devices due to their high strength and low weight.

- **D-LLM (4-bit quantized):** and/or Movies and other for purposes of building bridges, etcursos de worthwhiles Worthworth,ihu, huar Hore, and so on.

## Example 3

**Instruction:** Tell me about alpacas.
**QTALE (4-bit quantized):** Alpacas are a domesticated camelid species from the Andes. They are small, gentle, and valued for meat and dairy.
**D-LLM (4-bit quantized):** Alpacas are South American camelids raised for wool. They have soft fur, gentle temperament, and are easily trained.

## Example 4

**Instruction:** List all Canadian provinces in alphabetical order.
**QTALE (4-bit quantized):** Alberta, British Columbia, Manitoba, New Brunswick, Newfoundland, Nova Scotia, Ontario, Prince Edward Island, Quebec, Saskatchewan, Yukon, Northwest Territories.
**D-LLM (4-bit quantized):** Alberta, British Columbia, Manitoba, New Brunswick, Newfoundland, Nova Scotia, Ontario, Quebec, Saskatchewan, Yukon.

## Example 5

**Instruction:** Write a Python program that prints the first 10 Fibonacci numbers.
**QTALE (4-bit quantized):**

```
import math

def fibonacci(n):
    if n == 0:
        return 0
    elif n == 1:
        return 1
    else:
        return fibonacci(n-1) + fibonacci(n-2)
```

**D-LLM (4-bit quantized):**

```
for i in range(11):
    print(i)
```

