# OpenReview forum: "QTALE: Quantization-Robust Token-Adaptive Layer Execution for LLMs"
_ICML.cc/2026/Conference — ICML 2026 regular_

### Official Review · Reviewer_y6Sq · 2026-03-12

**Soundness:** 1
**Presentation:** 3
**Significance:** 2
**Originality:** 2
**Overall Recommendation:** 3
**Confidence:** 4

**Summary:**

This paper follows the research line of router-based layer skipping and explores how to integrate it with quantization. It argues that existing router-based skipping methods suffer from a router collapse problem, where only a subset of layers are sufficiently trained, reducing the redundancy needed for effective quantization. To address this, the paper introduces an entropy-based loss regularization to encourage greater training-path diversity, thereby improving robustness under quantization. In addition, it modifies the router output so that the execution ratio can be adjusted at inference time to restore redundancy when needed. Experiments demonstrate improvements over the DLLM baseline, achieving substantial memory savings through quantization while maintaining comparable latency speedups.

**Compliance With Llm Reviewing Policy:**

Affirmed.

**Final Justification:**

The rebuttal answered my concerns. However, I still feel that the method is limited and only applicable to D-LLM-like methods, with a narrow scope for quantization. Therefore, I would like to view the paper as borderline rather than a clear accept.

**Key Questions For Authors:**

1. What is the cost associated with adjusting the execution threshold? Is this step necessary in practice? Additionally, if this adjustment is not performed, how does the resulting accuracy compare to D-LLM?

2. Can you provide further analysis on how the proposed method improves robustness to quantization? Are there any direct indicators or empirical signals that demonstrate this effect?

**Limitations:**

This paper does not include the Impact Statements section.

**Strengths And Weaknesses:**

**Strengths**

1. The experiment section is solid, with experiment results on many benchmarks and settings (bitwidth, model type, compression techniques, etc.), demonstrating the improvement of the proposed method compared to the D-LLM baseline.

2. The motivation, the router collapsed problem, behind the proposed method is clear, and the proposed method directly addresses this problem, leading to accuracy gains. The overall reasoning from motivation to solution to empirical gains is coherent and well presented.

**Weaknesses**

1. The proposed method appears to be incremental and resembles an engineering refinement rather than a fundamentally new approach. First, the router collapse issue can be mitigated in various ways (e.g., temperature scaling), and the proposed entropy-based loss essentially acts as an additional regularization term rather than directly addressing the root cause of the problem. Moreover, the introduction of the new hyperparameter $\lambda_2$ requires careful tuning, which may limit the practicality of the method. Second, the paper introduces an execution-ratio adjustment mechanism by modifying the router output. However, similar functionality could arguably be achieved using the original router outputs by applying a softmax over the two logits (e.g., $s_{\text{exec}}$ and $s_{\text{skip}}$) and adjusting a threshold on the execution probability. Therefore, the ratio adjustment mechanism may not constitute a substantive contribution, leaving the entropy-based regularization as the primary novelty, which appears relatively incremental.

2. The empirical improvements reported in the experiments appear relatively modest and come at the cost of increased latency, as shown in Table 3. Given this trade-off, it is somewhat unclear whether the proposed method provides sufficient practical benefits to justify its additional complexity. Also, the method requires an additional execution-threshold search, typically performed via grid search. This process can be time-consuming and may need to be repeated for different tasks, which potentially reduces the practicality and generalizability of the method.

3. The paper argues that increasing training-path redundancy improves robustness to quantization, but the underlying rationale is not sufficiently explained. It remains unclear why increasing training-path redundancy specifically benefits quantization. For example, does it reduce activation outliers or mitigate quantization error in some measurable way? As currently presented, training-path redundancy seems like a general regularization effect that could benefit multiple aspects of model behavior, making it difficult to attribute the observed gains specifically to improved quantization robustness.

---

> ### Author Rebuttal · Authors · 2026-03-31
>
> We sincerely thank the reviewer for the valuable comments. Below, we provide detailed responses to each of the raised concerns and questions.
>
> **W1. Response to 'Incremental Contribution' Concern**
>
> *1. Overall Contribution*
>
> We respectfully disagree with the characterization of our method as an engineering refinement. Our contribution is to identify and address a previously overlooked failure mode when combining token-adaptive layer execution with quantization.
>
> Quantization alters router decisions and activates execution paths that are rarely explored during training, leading to accuracy degradation due to poor generalization over unseen paths. This issue is distinct from standard router collapse and is not addressed by existing methods.
>
> To address this, we propose a unified pipeline:
> * Entropy regularization: maintains router logits in a regime where Gumbel noise induces decision flipping, increasing path diversity during training.
> * Execution ratio adjustment: recovers accuracy under quantization by compensating for routing distribution shifts.
>
> These components are complementary. Entropy regularization improves training-time path coverage, while execution ratio adjustment addresses inference-time shifts, yielding consistent gains across models and tasks.
>
>
> *2. Hyperparameter Sensitivity*
>
> As shown in Figure 7 (Appendix A.2), the entropy regularization coefficient ($\lambda_2$) is not highly sensitive. Stable performance is achieved within a small range (e.g., 0.005–0.01), and we use a fixed default value across all experiments. While excessively large values may degrade accuracy, the valid range is sufficiently broad, resulting in minimal tuning overhead.
>
> *3. Thresholding Mechanism*
>
> We agree that thresholding itself is simple. However, to the best of our knowledge, this is the first work to demonstrate that adjusting the threshold can mitigate quantization-induced accuracy degradation.
>
> Moreover, as shown in Table 4, threshold adjustment alone provides limited benefit, whereas combining it with entropy regularization consistently improves performance. For example, on LLaMA2-7B with the CSQA benchmark, accuracy improves from 77.32 (baseline) to 78.48 with threshold adjustment alone, and further to 79.18 when combined with entropy regularization (78.86 without thresholding).
>
> This demonstrates that the key contribution lies in the joint design of training and inference strategies, rather than either component in isolation.
>
> **W2/Q1. Effectiveness of Execution Threshold Adjustment**
>
> *1. Clarification of the latency–accuracy trade-off*
>
> According to Table 3, our method maintains similar latency to D-LLM (18.2s vs. 17.8s) while achieving comparable speedup over the full model (1.26× vs. 1.28×). Despite this minimal overhead, it consistently improves accuracy. For example, on the CSQA benchmark, accuracy improves from 78.77 to 79.18 on LLaMA2-7B (Table 1), and from 73.96 to 78.41 on LLaMA3.2-3B (Table 2).
>
> *2. Ablation Study*
>
> The ablation study in Table 4 highlights the individual contributions of entropy regularization and execution ratio adjustment. The first row corresponds to D-LLM ($L_{entropy}$ = x, $\theta=0.5$), while the third row shows results with entropy regularization only ($L_{entropy}$ = o, $\theta=0.5$). Even without threshold adjustment, entropy regularization alone yields notable improvements. For example, on LLaMA2-7B with CSQA, accuracy improves from 77.32 to 78.86.
>
> *3. Calibration Cost*
>
> Threshold adjustment is a post-training procedure that uses a small calibration set of 300 samples. As discussed in Appendix A.4.4 (Table 12), a simple grid search is sufficient and requires 808.78 seconds (approximately 13.5 minutes).
>
> Importantly, threshold adjustment is optional in practice. If higher accuracy is required and a slight latency increase is acceptable, it can be applied. Otherwise, it can be skipped without affecting the core benefits of our method. This flexibility allows the approach to adapt to different deployment scenarios.
>
> **W3/Q2. Evidence on the Improvement of Quantization Robustness**
>
> Table 8 (Appendix A.3) shows that quantization changes token-path assignments by up to 40%, indicating significant routing shifts. In D-LLM, some layers become effectively pruned during training and are rarely updated (Figure 4). When quantization activates these under-trained paths, accuracy degrades.
>
> Therefore, our entropy-based training reduces the gap between router logits (Figure 4), thereby increasing the likelihood of decision flipping induced by Gumbel noise (Figure 5). This encourages a broader set of execution paths to be explored during training. As a result, the model learns more diverse routing behaviors, improving robustness to quantization-induced path shifts.
>
> Additional evidence can be found in the flipping ratio analysis for pruned layers (16–31), as discussed in our response to reviewer *1UFt - W1. Analysis of Impact of Entropy Regularization across Layers*.

---

> > ### Author Rebuttal · Reviewer_y6Sq · 2026-04-02
> >
> > My concerns are addressed. I raised my score.

---

> > > ### Author Response · Authors · 2026-04-03
> > >
> > > Thank you for updating your score and for confirming that your concerns have been addressed.
> > >
> > > We are glad that the rebuttal helped resolve the issues you raised.
> > > We would appreciate it if you could share whether there are any remaining concerns or other factors that led you to maintain a weak reject score.
> > >
> > > This would greatly help us better understand your evaluation.

---

### Official Review · Reviewer_RvGE · 2026-03-12

**Soundness:** 3
**Presentation:** 4
**Significance:** 3
**Originality:** 3
**Overall Recommendation:** 4
**Confidence:** 3

**Summary:**

This paper proposes QTALE (Quantization-Robust Token-Adaptive Layer Execution), a framework designed to seamlessly integrate token-adaptive layer execution (for FLOPs reduction) with post-training quantization (for memory footprint reduction) in Large Language Models (LLMs). The authors identify that naively combining these two efficiency techniques exacerbates accuracy degradation due to a compounding loss of redundancy—specifically, reduced training-path diversity and fewer active parameters during inference. To mitigate this, QTALE introduces two key contributions. First, it employs a quantization-robust training strategy using an entropy regularization term on the router outputs to encourage the exploration of diverse execution paths during fine-tuning. Second, it introduces a post-training, threshold-based execution ratio adjustment mechanism that allows flexible reintroduction of redundancy at inference time to absorb quantization errors. Comprehensive evaluations on LLaMA-family models (LLaMA2 and LLaMA3 variants) demonstrate that QTALE maintains comparable accuracy to quantization-only baselines (e.g., keeping the performance gap within 0.5% on CommonsenseQA benchmarks) while achieving simultaneous reductions in both computational cost and memory usage.

**Compliance With Llm Reviewing Policy:**

Affirmed.

**Final Justification:**

The authors have adequately addressed my previous concerns. While the method relies heavily on engineering optimizations, it still demonstrates sufficient novelty and potential. I recommend acceptance with minor revisions.

**Key Questions For Authors:**

1. **Throughput at Large Batch Sizes:** In Table 3, inference latency is evaluated at a very small batch size of 4. In production environments, LLMs are served with much larger batch sizes. How does dynamic token-adaptive routing impact memory bandwidth and GPU utilization when the batch size scales up (e.g., to 32, 64, or 128)? Does the fragmented execution path across tokens negate the theoretical FLOPs reduction in terms of actual wall-clock throughput?
2. **Hardware Implementation Details:** Did you implement custom CUDA kernels (e.g., using Triton) to handle the dynamic sub-layer skipping efficiently? Standard frameworks usually struggle to materialize theoretical FLOPs savings into actual latency reductions when control flow diverges per token.
3. **Robustness on Complex Reasoning Tasks:** The current evaluation relies heavily on CSQA and MMLU. Could the authors provide results on tasks requiring multi-step reasoning (e.g., GSM8K, MATH) or exact structural generation (e.g., HumanEval or MBPP)? These tasks are far more susceptible to the compounded errors from quantization and layer-skipping, and evaluating them would significantly strengthen the paper's claims on robustness.

**Limitations:**

yes

**Strengths And Weaknesses:**

**Strengths:**
1. **Practical Relevance:** The paper tackles a critical deployment bottleneck by unifying token-adaptive layer execution (for FLOPs reduction) and post-training quantization (for memory footprint reduction).
2. **Elegant Methodology:** The proposed solutions—an entropy regularization term ($\mathcal{L}_{entropy}$) during fine-tuning and a training-free threshold ($\theta$) for inference—are lightweight, intuitive, and avoid heavy architectural modifications.
3. **Solid Baselines:** Extensive evaluation on LLaMA-family models (2-7B, 3.1-8B, 3.2-3B) demonstrates that QTALE effectively closes the performance gap to quantization-only baselines.

**Weaknesses:**
1. **Discrepancy Between Theoretical FLOPs and Real-World Latency:** The theoretical FLOPs reduction (~46%) does not translate well to wall-clock speedup. In Table 3, quantized QTALE sometimes exhibits higher latency than the unquantized full model.
2. **Lack of Hardware/System-Level Details:** The evaluation uses a minimal batch size of 4. There is insufficient discussion on how token-level dynamic routing avoids severe memory fragmentation and poor GPU utilization at production-scale batch sizes.
3. **Missing Complex Reasoning Benchmarks:** The evaluation heavily relies on multiple-choice commonsense reasoning (CSQA) and MMLU, which are relatively robust to minor architectural perturbations. It lacks tasks requiring rigorous multi-step reasoning or structural generation, which are typically highly sensitive to layer-skipping and quantization errors.

---

> ### Author Rebuttal · Authors · 2026-03-31
>
> We sincerely thank the reviewer for the valuable comments. Below, we provide detailed responses to each of the raised concerns and questions.
>
> **Q1/W1/W2. Latency(Throughput) at Large Batch Sizes**
>
> First, we respectfully clarify that QTALE does not exhibit higher latency than the 16-bit full model. In Table 3, QTALE consistently achieves lower latency than the full-precision baseline across all settings.
>
> To examine the effect of batch size, we conduct additional experiments with larger batch sizes (8, 16, and 32). Please note that batch sizes ≥ 64 could not be supported on our A6000 GPU due to memory limitations.
>
> *Table RvGE-1.* Inference latency and relative speedup across batch sizes on A6000.
>
> | Bits | Method | Batch 8 | Batch 16 | Batch 32 |
> |----------|--------|--------|--------|--------|
> | 16-bit | Full | 22.9 (1.00x) | 24.0 (1.00x) | 25.8 (1.00x) |
> |        | D-LLM | 16.7 (1.37x) | 16.9 (1.42x) | 17.9 (1.44x) |
> |        | **QTALE** | **16.7 (1.37x)** | **17.0 (1.41x)** | **18.2 (1.42x)** |
> | 4-bit  | Full | 24.6 (0.93x) | 26.2 (0.92x) | 29.3 (0.88x) |
> |        | D-LLM | 17.5 (1.31x) | 18.3 (1.31x) | 19.6 (1.31x) |
> |        | **QTALE** | **18.0 (1.27x)** | **19.1 (1.25x)** | **19.9 (1.29x)** |
>
> The table above reports latency (in seconds), with speedup over the 16-bit full baseline shown in parentheses. All results are obtained under the same evaluation settings as in Table 3 of the manuscript. Specifically, we measure the total processing time for the benchmark tasks in Table 3 and report the average latency.
>
> QTALE consistently achieves lower latency than the full model across all batch sizes, demonstrating stable practical speedups.
> To mitigate fragmented execution, our implementation identifies active tokens based on routing decisions and gathers them to form dense inputs before each layer execution. This avoids per-token execution and maintains batched computation over active tokens, making the speedup robust to batch size.
>
> (Additional comments on the experimental setting) Latency increases as batch size grows, primarily because static batching is used for evaluation, which causes shorter sequences to wait for longer ones by padding with dummy tokens. To ensure a fair latency comparison, padding tokens are treated consistently across methods: they are fully processed in the full model, while they are dynamically skipped in D-LLM and QTALE.
>
> **Q2. Hardware Implementation Details**
>
> We implement custom Triton kernels to support dynamic sub-layer skipping. However, our current implementation only partially optimizes execution and does not fully exploit the theoretical efficiency of dynamic routing.
> The router layer consists of a linear projection followed by normalization and activation, and then routing decision computation. In our implementation, we apply kernel fusion only to the normalization and activation operations, along with a custom kernel for routing decision computation.
>
> As correctly pointed out by the reviewer, the achieved speedup does not fully match the theoretical FLOPs reduction due to remaining inefficiencies in kernel execution and control flow.
> We did not further optimize the kernels, as we believe that the current implementation is  sufficient to demonstrate practical latency improvements over full-model execution. Nevertheless, further optimization of the router block, including more aggressive kernel fusion, remains an important direction for future work toward maximizing hardware efficiency.
>
> **Q3/W3. Robustness on Complex Reasoning Tasks**
>
> We evaluate zero-shot accuracy of Llama3.2-3B and Llama3.1-8B on the GSM8K benchmark using input prompts for Chain-of-Thought (CoT) reasoning. We use the same hyperparameter settings as in our previous experiments (e.g., $\lambda_2 = 0.01$). While we use 300 samples for threshold calibration in other benchmarks, we use 64 samples for GSM8K. This is because CoT reasoning produces longer output sequences, and we adjust the calibration size to maintain a comparable calibration overhead across benchmarks.
>
>
> *Table RvGE-2.* GSM8K Accuracy (Acc.)
>
> | Bits | Method  | Acc. (%) ↑ |
> |--------|:------|---------|
> |      |**Llama3.2-3B** |
> | 16-bit | Full  | 30.50 |
> |        | D-LLM | 26.38 |
> |        | QTALE | **26.54** |
> | 4-bit  | Full  | 20.39 |
> |        | D-LLM | 16.98 |
> |        | QTALE | **23.28** |
> |          |**Llama3.1-8B** |
> | 16-bit | Full  | 45.26 |
> |        | D-LLM | 41.85 |
> |        | QTALE | **45.03** |
> | 4-bit  | Full  | 35.33 |
> |        | D-LLM | 32.60 |
> |        | QTALE | **42.83** |
>
> The GSM8K results show that QTALE consistently achieves higher accuracy than D-LLM after quantization. For example, the accuracy of 4-bit D-LLM and QTALE is 16.98 and 23.28, respectively, for Llama3.2-3B. These results demonstrate that our method remains effective on mathematical reasoning tasks.
>
> Thank you again for the feedback. We will incorporate our rebuttal discussions into the revised manuscript.

---

> > ### Author Rebuttal · Reviewer_RvGE · 2026-04-03
> >
> > Thank you very much for your rebuttal and for all the effort you have put into it. I will maintain my score.

---

> > > ### Author Response · Authors · 2026-04-03
> > >
> > > Thank you for your thoughtful review and for taking the time to consider our rebuttal.
> > >
> > > We sincerely appreciate your feedback.

---

### Official Review · Reviewer_NFQK · 2026-03-13

**Soundness:** 3
**Presentation:** 3
**Significance:** 3
**Originality:** 3
**Overall Recommendation:** 4
**Confidence:** 4

**Summary:**

This paper proposes a layer-skipping model based on quantized LLMs, which can more seamlessly integrate compute-saving token-adaptive layer execution (layer skipping) with memory-saving low-bit quantization. By introducing a quantization-robust training strategy to ensure training-path diversity and a flexible inference-time execution ratio adjustment mechanism, it overcomes the severe accuracy degradation caused by the compounded loss of model redundancy when these two techniques are naively combined.

**Compliance With Llm Reviewing Policy:**

Affirmed.

**Final Justification:**

The rebuttal addressed my main concerns, and I decide to keep my positive score.

**Key Questions For Authors:**

See weakness.

**Limitations:**

yes

**Strengths And Weaknesses:**

- Strength
    - **Feasible Integration:** The paper proposes a highly feasible and practical solution for executing token-adaptive layer skipping on quantized models, effectively mitigating the severe performance degradation seen in naive combinations.
    - **Thorough Experimental Validation:** The proposed framework is rigorously evaluated across multiple LLMs (e.g., LLaMA2, LLaMA3.1, LLaMA3.2) and diverse benchmarks, providing comprehensive and convincing experimental validation.
- Weakness
    - **Insufficient empirical justification for the core motivation:** The central premise of the paper is that additional parameter layers must be activated to compensate for the redundancy lost due to quantization. However, this motivation demands a more comprehensive demonstration through detailed ablation studies. The absence of a clear, quantitative correlation between varying quantization precisions and the necessary number of active parameter layers weakens the persuasiveness of the proposed mechanism.
    - **The current manuscript only provides the routing behavior for a fraction of the layers (e.g., layers 20, 23, 26).** Are there more comprehensive visualizations or global statistical summaries available for all layers to prove that the proposed method consistently prevents routing collapse across the entire network?
    - **Minor Formatting Issue:** The spacing between captions and the main text is insufficient, which slightly degrades the visual flow and reading experience.

---

> ### Author Rebuttal · Authors · 2026-03-31
>
> We sincerely thank the reviewer for the valuable comments. Below, we provide detailed responses to each of the raised concerns.
>
> **W1. Justification for the Core Motivation**
>
> QTALE addresses quantization robustness through two components: 1) entropy regularization to increase routing diversity during training, and 2) execution ratio adjustment to compensate for quantization-induced errors at inference. Below, we provide empirical justification for both components and their connection to quantization.
>
> *1) Entropy Regularization: Quantization-Induced Routing Shift*
>
> Table 8 (Appendix A.3) shows that quantization changes token-path assignments by up to 40%, indicating significant routing shifts. In D-LLM, some layers become effectively pruned during training and are rarely updated (Figure 4). When quantization activates these under-trained paths, accuracy degrades.
>
> Therefore, our entropy-based training reduces the gap between router logits (Figure 4), thereby increasing the likelihood of decision flipping induced by Gumbel noise (Figure 5). This encourages a broader set of execution paths to be explored during training. As a result, the model learns more diverse routing behaviors, improving robustness to quantization-induced path shifts.
>
> *2) Execution Ratio Adjustment: Compensation for Quantization Error*
>
> We can define the total error ($e_{total}$) as the sum of the error introduced by layer skipping ($e_{skip}$) and quantization ($e_{quant}$) as follows:
>
> $$e_{total} = e_{skip} + e_{quant}$$
>
> Quantization increases $e_{quant}$, while layer skipping introduces $e_{skip}$. As quantization becomes more aggressive, the combined error increases. Execution ratio adjustment compensates for this by increasing the number of executed layers, thereby reducing $e_{skip}$ and balancing the increase in $e_{quant}$.
>
> To empirically validate this decomposition, we measure the cosine similarity between the dense FP16 model output and the outputs of models with quantization and/or layer skipping. The experiment is conducted under 3-bit quantization using 150 samples from the ARC-e dataset, where we explicitly control layer skipping by forcing the model to skip the 23rd layer.
>
> *Table NFQK-1.* Cosine similarity(Cos.) between quantized/layer skipped model output and the dense FP16 output
>
> | Quant.| Skip Layer | Cos.|
> |:-:|:-:|:-:|
> |x|o|0.9610|
> |o|o|0.8219|
> |o|x|0.8605|
>
> Under quantization, executing the layer yields higher similarity to the dense model (0.8605) than skipping the layer (0.8219). This indicates that skipping introduces additional deviation, while executing the layer reduces this error.
>
> This observation is consistent with our formulation: skipping increases $e_{skip}$, whereas executing reduces it, thereby lowering $e_{total}$. As a result, increasing the execution ratio effectively compensates for quantization-induced error, leading to outputs that better align with the dense model.
>
> **W2. Routing Behavior across the Entire Network**
>
> *Table 1UFt-2.* Gumbel noise–induced execution decision flipping ratio of Llama3.1-8B (Ours/D-LLM)
>
> || Epoch 6|Epoch 7|Epoch 8|Epoch 9|
> |-|-|-|-|-|
> |arce|**0.0459** / 0.0246|**0.0365** / 0.0338|**0.0425** / 0.0330|**0.0429** / 0.0327|
> |boolq|**0.0288** / 0.0153|**0.0331** / 0.0149|**0.0326** / 0.0136|**0.0321** / 0.0135|
> |piqa|**0.0236** / 0.0181|**0.0261** / 0.0158|**0.0238** / 0.0162|**0.0239** / 0.0145|
> |winogra.|**0.0307** / 0.0178|**0.0325**/ 0.0164|**0.0337** / 0.0137|**0.0342** / 0.0143|
>
> To clearly validate the impact of entropy regularization on the routing collapse across the network, we analyze the average flipping ratio of router decisions for layers 16–31 at later training stages (epochs 6–9) for both D-LLM and our method, as shown in the table above.
>
> We focus on these layers and training stages because most layers are consistently executed in early layers or during the initial phase of training (Figure 2 of the manuscript). Entropy regularization is designed to maintain router entropy in a regime that enables decision flipping during training by regulating the execution and bypass logits within a range where Gumbel noise can flip the decision. This prevents premature pruning of unexecuted paths. Therefore, its effect is best examined in deeper layers and later training stages, where path-pruning behavior is more likely to occur.
>
> Our method maintains consistently higher flipping ratios across layers 16–31 during epochs 6–9. It indicates that router decisions remain active rather than collapsing to a single outcome. This suggests that our approach mitigates the path pruning issue observed in D-LLM through entropy regularization, thereby preserving training-path diversity.
>
>
> **W3. Minor Formatting Issue**
>
> We thank the reviewer for pointing this out. We will ensure to increase the spacing between the captions and the main text in the revised manuscript.
>
> Thank you again for the feedback. We will incorporate our rebuttal discussions into the revised manuscript.

---

> > ### Author Rebuttal · Reviewer_NFQK · 2026-04-03
> >
> > Thank you for your detailed rebuttal, I maintain my score.

---

> > > ### Author Response · Authors · 2026-04-03
> > >
> > > Thank you for your thoughtful review and for taking the time to consider our rebuttal.
> > >
> > > We sincerely appreciate your feedback.

---

### Official Review · Reviewer_1UFt · 2026-03-15

**Soundness:** 2
**Presentation:** 3
**Significance:** 3
**Originality:** 3
**Overall Recommendation:** 4
**Confidence:** 2

**Summary:**

This paper proposes a model combining layer skipping with quantization, and by introducing a robust training strategy and a dynamic redundancy compensation mechanism during inference, it resolves the severe accuracy collapse caused by the direct superposition of these two techniques.

**Compliance With Llm Reviewing Policy:**

Affirmed.

**Final Justification:**

My concerns have been addressed, I will maintain my score at 4.

**Key Questions For Authors:**

- Is the proposed method applicable to models with fine-grained routing, such as SkipGPT, which routes both attention and FFN layers?
- Dynamic routing is known to be difficult to train. Do the proposed components—Execution Ratio Adjustment and the entropy regularization loss—introduce additional training instability or require careful hyperparameter tuning in practice?

**Limitations:**

Yes

**Strengths And Weaknesses:**

**Strength**
- The paper tackles a underexplored bottleneck in LLM deployment that the compounded redundancy loss when naively combining layer skipping and quantization. This is a practical and contribution to the field of efficient LLMs.
- The experimental validation across multiple architectures is thorough, demonstrating robust performance retention even under aggressive compression.

**Weakness**
- While introducing entropy regularization is well-motivated, its empirical validation needs strengthening. To fully demonstrate its effectiveness, the authors should provide a global visualization of activations across all layers (rather than just a few) and explicitly compare the final execution ratios between QTALE and D-LLM to clarify its actual impact on path redundancy.
- Several **highly relevant works** on dynamic or adaptive computation in LLMs, such as SkipGPT and Informed Routing in LLMs, are not discussed. Including and comparing with these approaches would help better position the contribution within the broader literature on dynamic routing and layer skipping.

[1] SkipGPT: Dynamic Layer Pruning Reinvented with Token Awareness and Module Decoupling. ICML 2025.

[2] Informed Routing in LLMs: Smarter Token-Level Computation for Faster Inference. arxiv/2510.13831.

---

> ### Author Rebuttal · Authors · 2026-03-31
>
> We sincerely thank the reviewer for the valuable comments. Below, we provide detailed responses to each of the raised concerns and questions.
>
> **W1. Analysis of Impact of Entropy Regularization across Layers**
>
> *Table 1UFt-1.* Gumbel noise–induced execution decision flipping ratio of Llama3.1-8B (Ours/D-LLM)
>
> || Epoch 6       |    Epoch 7   | Epoch 8       | Epoch 9       |
> |------------|------------------|------------------|------------------|------------------|
> | arce       | **0.0459** / 0.0246  | **0.0365** / 0.0338  | **0.0425** / 0.0330  | **0.0429** / 0.0327  |
> | boolq    | **0.0288** / 0.0153  | **0.0331** / 0.0149  | **0.0326** / 0.0136  | **0.0321** / 0.0135  |
> | piqa       | **0.0236** / 0.0181  | **0.0261** / 0.0158  | **0.0238** / 0.0162  | **0.0239** / 0.0145  |
> | winogra. | **0.0307** / 0.0178  | **0.0325** / 0.0164 | **0.0337** / 0.0137 | **0.0342** / 0.0143  |
>
> To clearly validate the impact of entropy regularization across the network, we analyze the average flipping ratio of router decisions for layers 16–31 at later training stages (epochs 6–9) for both D-LLM and our method, as shown in the table above.
>
> We focus on these layers and training stages because most layers are consistently executed in early layers or during the initial phase of training (Figure 2 of the manuscript). Entropy regularization is designed to maintain router entropy in a regime that enables decision flipping during training by regulating the execution and bypass logits within a range where Gumbel noise can flip the decision. This prevents premature pruning of unexecuted paths. Therefore, its effect is best examined in deeper layers and later training stages, where path-pruning behavior is more likely to occur.
>
> Our method maintains consistently higher flipping ratios across layers 16–31 during epochs 6–9. It indicates that router decisions remain active rather than collapsing to a single outcome. This suggests that our approach mitigates the path pruning issue observed in D-LLM through entropy regularization, thereby preserving training-path diversity.
>
> **W2/Q1. Applicability of QTALE to Fine-Grained Routing Models**
>
> The proposed method is applicable to a broad class of fine-grained routing models that employ Gumbel-Softmax for training the router and argmax for inference-time routing decisions. This is because QTALE introduces entropy regularization on the Gumbel-Softmax outputs and adjusts the execution ratio by modifying the argmax decision.
>
> Since this training and inference scheme is commonly used in fine-grained routing models, QTALE can be readily applied to a wide range of such architectures. For example, SkipGPT adopts Gumbel-Softmax during training and argmax during inference, making the integration of QTALE straightforward.
>
> We attempted to conduct additional experiments on SkipGPT using the official implementation. However, we encountered environment configuration issues, and reproducing the full training pipeline was required for proper integration. As a result, it was difficult to include these results within the rebuttal timeline. Nevertheless, based on the architectural compatibility described above, we expect QTALE to be directly applicable to SkipGPT and similar models.
>
> We thank the reviewer for raising this important point, which highlights the broader applicability of our method.
>
> **Q2. Training Stability and Hyperparameter Sensitivity**
>
> We agree that dynamic routing is inherently difficult to train. However, we find that the proposed components do not introduce additional training difficulty for dynamic routing models.
>
> First, execution ratio adjustment is a post-training technique that calibrates the execution ratio after training. Therefore, it does not affect the training procedure.
>
> Next, for the entropy regularization loss, as demonstrated in Figure 7 (Appendix A.2) of the manuscript, the training performance is not highly sensitive to the entropy regularization coefficient ($\lambda_2$). We evaluate $\lambda_2$ in the range of [0.005, 0.5] and observe that values in the range of [0.005, 0.01] yield stable training behavior.
> However, we note that excessively large values of $\lambda_2$ can cause the entropy loss term to dominate the objective, leading to degraded accuracy. Therefore, it is important to keep $\lambda_2$ within a small range.
> For all other hyperparameters, we directly adopt the configurations from the D-LLM baseline.
>
> Thank you again for the feedback. We will incorporate our rebuttal discussions into the revised manuscript.

---

> > ### Author Rebuttal · Reviewer_1UFt · 2026-04-03
> >
> > Thank you for the detailed rebuttal.
> >
> > My concerns have been addressed, I will maintain my score.

---

> > > ### Author Response · Authors · 2026-04-03
> > >
> > > Thank you for your thoughtful review and for taking the time to consider our rebuttal.
> > >
> > > We sincerely appreciate your feedback.

---

### Decision · Program_Chairs · 2026-04-30

**Decision:**

Accept (regular)

**Comment:**

This paper proposes to integrate token-adaptive selection and quantization through a training strategy that actively explores diverse execution paths during fine-tuning, together with a post-training mechanism that allows flexible adjustment of the execution ratio at inference time so that redundancy can be reintroduced when needed.

The reviewers highlighted several strengths of the work, including the feasibility of the integration and the thorough experimental validation. Reviewer y6Sq noted in the final justification that the method is somewhat limited in scope and appears mainly applicable to D-LLM-like methods, with a relatively narrow setting for quantization. In my view, this is not a major weakness, but the experimental comparisons in the paper also seem to focus only on the D-LLM baseline, while token-adaptive methods more broadly include many other relevant approaches. The paper would therefore be stronger if the authors included a wider range of baselines for comparison.

I was also somewhat confused by the boldface formatting of the results tables. It is not very intuitive to bold results across all three models in the current way, and this makes it harder to identify which method actually performs best. The presentation of the paper, especially the experimental results, could be improved to make the comparisons clearer and more reader-friendly.

Overall, I think the paper receives generally positive scores of 4, 4, 4, 3 and has a meaningful idea and good strengths, but the presentation and experimental writing would benefit from further revision.